# Representation Power of Graph Convolutions : Neural Tangent Kernel Analysis

## Abstract

The fundamental principle of Graph Neural Networks (GNNs) is to exploit the structural information of the data by aggregating the neighboring nodes using a 'graph convolution'. Therefore, understanding its influence on the network performance is crucial. Convolutions based on graph Laplacian have emerged as the dominant choice with the symmetric normalization of the adjacency matrix $\mathbf{A}$, defined as $\mathbf{D}^{-\frac{1}{2}}\mathbf{A}\mathbf{D}^{-\frac{1}{2}}$, being the most widely adopted one, where $\mathbf{D}$ is the degree matrix. However, some empirical studies show that row normalization $\mathbf{D}^{-1}\mathbf{A}$ outperforms it in node classification. Despite the widespread use of GNNs, there is no rigorous theoretical study on the representation power of these convolution operators, that could explain this behavior. In this work, we analyze the influence of the graph convolutions theoretically using *Graph Neural Tangent Kernel* in a semi-supervised node classification setting. Under a *Degree Corrected Stochastic Block Model* we analyze different graphs that have homophilic, heterophilic and core-periphery structures, and prove that: (i) row normalization preserves the underlying class structure better than other convolutions; (ii) performance degrades with network depth due to over-smoothing, but the loss in class information is the slowest in row normalization; (iii) skip connections retain the class information even at infinite depth, thereby eliminating over-smoothing. We finally validate our theoretical findings numerically and on real datasets.

## 1 Introduction

With the advent of Graph Neural Networks (GNNs), there has been a tremendous progress in the development of computationally efficient state-of-the-art methods in various graph based tasks, including drug discovery, community detection and recommendation systems (Wieder et al., 2020; Fortunato & Hric, 2016; van den Berg et al., 2017). Many of these problems depend on the structural information of the entities along with the features for effective learning. Because GNNs exploit this topological information encoded in the graph, it can learn better representation of the nodes or the entire graph than traditional deep learning techniques, thereby achieving state-of-the-art performances. In order to accomplish this, GNNs apply aggregation function to each node in a graph that combines the features of the neighboring nodes, and its variants differ principally in the methods of aggregation. For instance, graph convolution networks use mean neighborhood aggregation through spectral approaches (Bruna et al., 2014; Defferrard et al., 2016; Kipf & Welling, 2017) or spatial approaches (Hamilton et al., 2017; Duvenaud et al., 2015; Xu et al., 2019), graph attention networks apply multi-head attention based aggregation (Velickovic et al., 2018) and graph recurrent networks employ complex computational module (Scarselli et al., 2008; Li et al., 2016). Of all the aggregation policies, the spectral approach based on graph Laplacian is most widely used in practice, specifically the one proposed by Kipf & Welling (2017) owing to its simplicity and empirical success. In this work, we focus on such graph Laplacian based aggregations in Graph Convolution Networks (GCNs), which we refer to as *graph convolutions* or *diffusion operators*.

Kipf & Welling (2017) propose a GCN for node classification, a semi-supervised task, where the goal is to predict the label of a node using its feature and neighboring node information. This work suggests symmetric normalization $\mathbf{S}_{sym} = \mathbf{D}^{-\frac{1}{2}}\mathbf{A}\mathbf{D}^{-\frac{1}{2}}$ as the graph convolution. Ever since its introduction, $\mathbf{S}_{sym}$ remains the popular choice. However, subsequent works (Wang et al., 2018; Wang & Leskovec, 2020; Ragesh et al., 2021) explore row normalization $\mathbf{S}_{row} = \mathbf{D}^{-1}\mathbf{A}$ and particu-

larly, Wang et al. (2018) observes that $\mathbf{S}_{row}$ outperforms $\mathbf{S}_{sym}$ for two-layered GCN empirically. Intrigued by this observation, and as both $\mathbf{S}_{sym}$ and $\mathbf{S}_{row}$ are simply degree normalized adjacency matrices, we study the behavior over depth and observe that $\mathbf{S}_{row}$ performs better than $\mathbf{S}_{sym}$ in this case as well, as illustrated in Figure 1 (Details of the experiment in Appendix B.1).

Furthermore, another striking observation from Figure 1 is that the performance of GCN without skip connections decreases considerably with depth for both $\mathbf{S}_{sym}$ and $\mathbf{S}_{row}$. This contradicts the conventional wisdom about standard neural networks which exhibit improvement in the performance as depth increases. Several works (Kipf & Welling, 2017; Chen et al., 2018; Wu et al., 2019) observe this behavior empirically and attribute it to the over-smoothing effect from the repeated application of the diffusion operator, resulting in averaging out of the feature information to a degree where it becomes uninformative (Li et al., 2018; Oono & Suzuki, 2019; Esser et al., 2021). As a solution to this problem, Chen et al. (2020) and Kipf & Welling (2017) propose different forms of skip connections that overcome the smoothing effect and thus outperform the vanilla

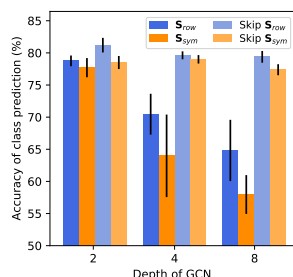

Figure 1: Performance of GCN over depth with and without skip connections using $\mathbf{S}_{sym}$ and $\mathbf{S}_{row}$ evaluated on Cora dataset.

GCN. Extending it to the comparison of graph convolutions, our experiment shows $\mathbf{S}_{row}$ is preferable to $\mathbf{S}_{sym}$ over depth even in GCNs with skip connections (Figure 1). Naturally, we ask: *what characteristics of $\mathbf{S}_{row}$ enable better representation learning than $\mathbf{S}_{sym}$ in GCNs?*

Rigorous theoretical analysis is particularly challenging in GCNs compared to the standard neural networks because of the graph convolution. Adding skip connections further increase the complexity of the analysis. To overcome these difficulties, we consider GCN in infinite width limit wherein the *Neural Tangent Kernel (NTK)* captures the network characteristics very well (Jacot et al., 2018). The infinite width assumption is not restrictive for graph convolution analysis as the convolution operates on the graph and not the network directly, thus showing same observations as trained GCN (Figure 5). Moreover, NTK enables the analysis to be parameter-free and hence eliminating additional complexity induced for example by optimization. Through the lens of NTK, we study the impact of different graph convolutions under a specific data distributional assumption — *Degree Corrected Stochastic Block Model (DC-SBM)*(Karrer & Newman, 2011), a sparse random graph model. The node degree heterogeneity induced in DC-SBM allows us to analyze the effect of different types of normalization of the adjacency matrix, thus revealing the characteristic difference between $\mathbf{S}_{sym}$ and $\mathbf{S}_{row}$. Additionally, this model enables analysis of graphs that have *homophilic, heterophilic and core-periphery* structures. In this paper, we present a formal approach to analyze GCNs and, specifically, *the representation power of different graph convolutions, the influence of depth and the role of skip connections.* This is a significant step toward understanding GCNs as it facilitates for more informed network design choices like the convolution and depth, as well as development of more competitive methods based on grounded theoretical reasoning rather than heuristics.

**Contributions.** This paper provides rigorous theoretical analysis of the discussed empirical observations in GCN under DC-SBM distribution using graph NTK, leading to the following contributions.

**(i)** In Section 2, we derive the NTK for GCN in infinite width limit considering node classification setting. Using the NTK for linear GCN and under DC-SBM data distribution, we show in Section 3 that $\mathbf{S}_{row}$ preserves class information by computing the population NTK for different graph convolutions. We also present numerical validation of the result in homophilic and heterophilic graphs.

**(ii)** We prove the convolution operator specific over-smoothing effect in vanilla GCN by showing the degradation in class separability with depth in Section 3.1, and also illustrate it experimentally.

**(iii)** In Section 4, we leverage the power of NTK to analyze two different skip connections (Kipf & Welling, 2017; Chen et al., 2020). We derive the corresponding NTKs and show that skip connections retain class information even at infinite depth along with numerical validation.

**(iv)** We show that $\mathbf{S}_{sym}$ maybe preferred over $\mathbf{S}_{row}$ in absence of class structure in Section 5, and validate the theoretical results on real datasets *Cora* in Section 6 and *Citeseer* in Appendix B.5.

We finally conclude in Section 7 with the discussion on the impact of the result and further possibilities, and provide all the proofs, experimental details and additional experiments in the appendix.

**Related Work.** While GNNs are extensively used in practice, their understanding is limited, and the analysis is mostly restricted to empirical approaches (Bojchevski et al., 2018; Zhang et al., 2018; Ying et al., 2018; Wu et al., 2020). Beyond empirical methods, rigorous theoretical analysis using *learning theoretical bounds* such as VC Dimension, Scarselli et al. (2018) or PAC-Bayes Liao et al. (2021) are propounded. Rademacher Complexity bounds (Garg et al., 2020; Esser et al., 2021) show that normalized graph convolution is beneficial, but those works do not provide insight on the different normalizations, and their influence on the GCN performance. Another possible tool is the NTK using which interesting theoretical insights in deep neural networks are derived (e.g. (Du et al., 2019a)). In the context of GNNs, Du et al. (2019b) derives the NTK in the supervised setting (each graph is a data instance to be classified) and empirically studies the NTK performance, however does not extend it to a theoretical analysis. In contrast, we derive the NTK in the *semi-supervised* setting for GCN with and without skip connections, and use it to further theoretically analyze the influence of different convolutions with respect to over-smoothing. Theoretical studies (Oono & Suzuki, 2019; Cai & Wang, 2020) show that over-smoothing causes the expressive power of GNNs to decrease exponentially with depth, while Keriven (2022) proves that in linear GNNs a finite number of convolutions improves learning before over-smoothing kicks in. While over-smoothing and role of skip connections in GNNs are theoretically analyzed in some works (Esser et al., 2021), the influence of different convolutions that causes over-smoothing and their interplay with skip connections is not studied. For a comprehensive theory survey see Jegelka (2022).

**Notations.** We represent matrix and vector by bold faced uppercase and lowercase letters, respectively, the matrix Hadamard (entry-wise) product by $\odot$ and the scalar product by $\langle .,. \rangle$. We use $\mathbf{M}^{\odot k}$ to denote Hadamard product of matrix $\mathbf{M}$ with itself repeated $k$ times. Let $\mathcal{N}(\mu, \boldsymbol{\Sigma})$ be Gaussian distribution with mean $\mu$ and co-variance $\boldsymbol{\Sigma}$. We use $\dot{\sigma}(.)$ to represent derivative of function $\sigma(.)$, $\mathbf{1}_{n \times n}$ for the $n \times n$ matrix of ones, $\mathbf{I}_n$ for identity matrix of size $n \times n$, $\mathbb{1}[.]$ for indicator function, $\mathbb{E}[.]$ for expectation, and $[d] = \{1, 2, \ldots, d\}$.

## 2 NEURAL TANGENT KERNEL FOR GRAPH CONVOLUTIONAL NETWORK

Before going into a detailed analysis of graph convolutions we provide a brief background on *Neural Tangent Kernel* (NTK) and derive its formulation in the context of node level prediction using infinitely-wide GCNs. Jacot et al. (2018); Arora et al. (2019); Yang (2019) show that the behavior and generalization properties of randomly initialized wide neural networks trained by gradient descent with infinitesimally small learning rate is equivalent to a kernel machine. Furthermore, Jacot et al. (2018) also show that the change in the kernel during training decreases as the network width increases, and hence, asymptotically, one can represent an infinitely wide neural network by a deterministic NTK, which is defined by the gradient of the network with respect to its parameters as

$$\boldsymbol{\Theta}(\mathbf{x}, \mathbf{x}') := \underset{\mathbf{W} \sim \mathcal{N}(\mathbf{0}, \mathbf{I})}{\mathbb{E}} \left[ \left\langle \frac{\partial F(\mathbf{W}, \mathbf{x})}{\partial \mathbf{W}}, \frac{\partial F(\mathbf{W}, \mathbf{x}')}{\partial \mathbf{W}} \right\rangle \right]. \tag{1}$$

Here $F(\mathbf{W}, \mathbf{x})$ represents the output of the network at data point $\mathbf{x}$ parameterized by $\mathbf{W}$ and the expectation is with respect to $\mathbf{W}$, where all the parameters of the network are randomly sampled from Gaussian distribution. Although the 'infinite width' assumption is too strong to model real (finite width) neural networks, and the absolute performance may not exactly match, the empirical trends of NTK match the corresponding network counterpart, allowing us to draw insightful conclusions. This trade-off is worth considering as this allows the analysis of over-parameterised neural networks without having to consider hyper-parameter tuning and training.

**Formal GCN Setup and Graph NTK.** We present the formal setup of GCN and derive the corresponding NTK, using which we analyze different graph convolutions. Given a graph with $n$ nodes and a set of node features $\{\mathbf{x}_i\}_{i=1}^n \subset \mathbb{R}^f$, we may assume without loss of generality that the set of observed labels $\{\mathbf{y}_i\}_{i=1}^m$ correspond to first $m$ nodes. We consider $K$ classes, thus $\mathbf{y}_i \in \{0, 1\}^K$ and the goal is to predict the $n - m$ unknown labels $\{\mathbf{y}_i\}_{i=m+1}^n$. We represent the observed labels of $m$ nodes as $\mathbf{Y} \in \{0, 1\}^{m \times K}$, and the node features as $\mathbf{X} \in \mathbb{R}^{n \times f}$ with the assumption that entire $\mathbf{X}$ is available during training. We define $\mathbf{S} \in \mathbb{R}^{n \times n}$ to be the graph convolution operator as an expression of the adjacency matrix $\mathbf{A}$ and the degree matrix $\mathbf{D}$. The GCN of depth $d$ is given by

$$F_\mathbf{W}(\mathbf{X}, \mathbf{S}) := \sqrt{\frac{c_\sigma}{h_d}} \mathbf{S} \sigma \left( \ldots \sigma \left( \sqrt{\frac{c_\sigma}{h_1}} \mathbf{S} \sigma \left( \mathbf{S} \mathbf{X} \mathbf{W}_1 \right) \mathbf{W}_2 \right) \ldots \right) \mathbf{W}_{d+1} \tag{2}$$

where $\mathbf{W} := \{\mathbf{W}_i \in \mathbb{R}^{h_{i-1} \times h_i}\}_{i=1}^{d+1}$ is the set of learnable weight matrices with $h_0 = f$ and $h_{d+1} = K$, $h_i$ is the size of layer $i \in [d]$ and $\sigma : \mathbb{R} \to \mathbb{R}$ is the point-wise activation function. We initialize all the weights to be i.i.d $\mathcal{N}(0,1)$ and optimize it using gradient descent. We derive the NTK for the GCN in infinite width setting, that is, $h_1, \ldots, h_d \to \infty$. While this setup is similar to Kipf & Welling (2017), it is important to note that we consider linear output layer so that NTK remains constant during training (Liu et al., 2020) and additionally add a normalization $\sqrt{c_\sigma / h_i}$ for layer $i$ to ensure that the input norm is approximately preserved and $c_\sigma^{-1} = \underset{u \sim \mathcal{N}(0,1)}{\mathbb{E}} \left[ (\sigma(u))^2 \right]$ (similar to Du et al. (2019a)). The following theorem states the NTK between every pair of nodes, as a $n \times n$ matrix that can be computed at once, as shown below.

**Theorem 1 (NTK for Vanilla GCN)** *For the vanilla GCN defined in* (2)*, the NTK $\mathbf{\Theta}$ at depth $d$ is*

$$\mathbf{\Theta}^{(d)} = \sum_{k=1}^{d+1} \mathbf{\Sigma}_k \odot \left(\mathbf{SS}^T\right)^{\odot(d+1-k)} \odot \left( \bigodot_{k'=k}^{d+1-k} \dot{\mathbf{E}}_{k'} \right). \qquad (3)$$

*Here $\mathbf{\Sigma}_k \in \mathbb{R}^{n \times n}$ is the co-variance between nodes of layer $k$, and is given by $\mathbf{\Sigma}_1 = \mathbf{SXX}^T\mathbf{S}^T$, $\mathbf{\Sigma}_k = \mathbf{SE}_{k-1}\mathbf{S}^T$ with $\mathbf{E}_k = c_\sigma \underset{\mathbf{F} \sim \mathcal{N}(\mathbf{0}, \mathbf{\Sigma}_k)}{\mathbb{E}} \left[ \sigma(\mathbf{F})\sigma(\mathbf{F})^T \right]$ and $\dot{\mathbf{E}}_k = c_\sigma \underset{\mathbf{F} \sim \mathcal{N}(\mathbf{0}, \mathbf{\Sigma}_k)}{\mathbb{E}} \left[ \dot{\sigma}(\mathbf{F})\dot{\sigma}(\mathbf{F})^T \right].$*

**Comparison to Du et al. (2019b).** While the NTK in (3) is similar to the graph NTK in Du et al. (2019b), the main difference is that NTK in our case is computed for all pairs of nodes in a graph as we focus on semi-supervised node classification, whereas Du et al. (2019b) considers supervised graph classification where input is many graphs and so the NTK is evaluated for all pairs of graphs.

## 3 CONVOLUTION OPERATOR $\mathbf{S}_{row}$ PRESERVES CLASS INFORMATION

We use the derived NTK in Theorem 1 to analyze different graph convolutions for $\mathbf{S}$ defined in Definition 1 by making the following assumption on the network.

**Assumption 1 (Linear GCN with orthonormal features)** *GCN in* (2) *is said to be linear with orthonormal features if the activation function $\sigma(x) = x$ and $\mathbf{XX}^T = \mathbf{I}_n$.*

**Remark on Assumption 1.** The linear activation does not impact the performance of a GCN significantly as Wu et al. (2019) empirically demonstrates that the linearized GCN performance is at par with the non-linear models with much reduced complexity. Additional orthonormal features assumption eliminates the influence of the features and facilitates identification of the influence of different convolution operators. Besides, the evaluation of our theoretical results without this assumption on real datasets is presented in Section 6 and Appendix B.5 that substantiate our findings.

Therefore, the NTK for linear GCN with orthonormal features of depth $d$ is,

$$\mathbf{\Theta}^{(d)} = \sum_{k=1}^{d+1} \mathbf{\Sigma}_k \odot \left(\mathbf{SS}^T\right)^{\odot(d+1-k)} \text{ with } \mathbf{\Sigma}_k = \mathbf{S}^k {\mathbf{S}^k}^T. \qquad (4)$$

**Definition 1** *Symmetric degree normalized $\mathbf{S}_{sym} = \mathbf{D}^{-\frac{1}{2}}\mathbf{A}\mathbf{D}^{-\frac{1}{2}}$, row normalized $\mathbf{S}_{row} = \mathbf{D}^{-1}\mathbf{A}$, column normalized $\mathbf{S}_{col} = \mathbf{A}\mathbf{D}^{-1}$ and unnormalized $\mathbf{S}_{adj} = \frac{1}{n}\mathbf{A}$ convolutions.*

While the NTK in (4) gives a precise characterization of the infinitely wide GCN, we can not directly draw conclusions about the convolution operators without further assumptions on the input graph. Therefore, we consider a planted graph model as described below, that helps in establishing the exact representation power of each operator.

**Random Graph Model.** We consider that the underlying graph is from the Degree Corrected Stochastic Block Model (DC-SBM) (Karrer & Newman, 2011) since it enables us to distinguish between $\mathbf{S}_{sym}$, $\mathbf{S}_{row}$, $\mathbf{S}_{col}$ and $\mathbf{S}_{adj}$ by allowing non-uniform degree distribution on the nodes. The model is defined as follows: Consider a set of $n$ nodes divided into $K$ latent classes (or communities), $\mathcal{C}_i \in [1, K]$. The DC-SBM model is characterized by the parameters $p, q \in [0,1]$—governing the edge probabilities inside and outside classes—and the degree correction vector

$\boldsymbol{\pi} = (\pi_1, \ldots, \pi_n) \in [0, 1]^n$ with $\sum_i \pi_i = 1$. A random graph on $n$ nodes, generated from DC-SBM, has mutually independent edges with edge probabilities specified by the population adjacency matrix $\mathbf{M} = \mathbb{E}[\mathbf{A}] \in \mathbb{R}^{n \times n}$, where

$$\mathbf{M}_{ij} = \begin{cases} p\pi_i\pi_j & \text{if } \mathcal{C}_i = \mathcal{C}_j \\ q\pi_i\pi_j & \text{if } \mathcal{C}_i \neq \mathcal{C}_j \end{cases}$$

This allows us to model different graph types: **Homophilic graphs:** $0 \leq q < p \leq 1$, **Heterophilic graphs:** $0 \leq p < q \leq 1$ and **Core-Periphery graphs:** $p = q$ (no assumption on class structure) and $\boldsymbol{\pi}$ encode core and periphery. It is evident that the NTK is a complex quantity and computing its expectation is challenging given the dependency of terms from the degree normalization in $\mathbf{S}$, its powers $\mathbf{S}^i$ and $\mathbf{SS}^T$. To simplify our analysis, we make the following assumption on DC-SBM,

**Assumption 2 (Population DC-SBM)** *The graph has a weighted adjacency $\mathbf{A} = \mathbf{M}$.*

**Remark on Assumption 2.** Assuming $\mathbf{A} = \mathbf{M}$ is equivalent to analyzing DC-SBM in expected setting and it further enables the computation of analytic expression for the population NTK instead of the expected NTK. Moreover, we observe empirically that this analysis hold for random DC-SBM setting as well. In addition, this consideration also implies addition of self loop with a probability $p$.

In the following theorem, we state the population NTK for graph convolutions $\mathbf{S}_{sym}$, $\mathbf{S}_{row}$, $\mathbf{S}_{col}$ and $\mathbf{S}_{adj}$ for $K = 2$ with Assumption 1 and 2. The result extends to $K > 2$ as discussed in the appendix.

**Theorem 2 (Population NTKs $\tilde{\Theta}$ for the four graph convolutions S)** *Let Assumption 1 and 2 hold, $K = 2$ and $r = \frac{p-q}{p+q}$, $\delta_{ij} = (-1)^{\mathbb{1}[\mathcal{C}_i \neq \mathcal{C}_j]}$. Furthermore, $\boldsymbol{\pi}$ is chosen such that $\sum_{i=1}^n \pi_i \mathbb{1}[\mathcal{C}_i = k] = \frac{1}{K}$ and $\sum_{i=1}^n \pi_i^2 \mathbb{1}[\mathcal{C}_i = k] = \gamma \, \forall k$, where $\gamma$ is a constant. Then $\forall i, j$, population NTKs $\tilde{\Theta}_{sym}$, $\tilde{\Theta}_{row}$, $\tilde{\Theta}_{col}$ and $\tilde{\Theta}_{adj}$ of depth $d$ for $\mathbf{S} = \mathbf{S}_{sym}$, $\mathbf{S}_{row}$, $\mathbf{S}_{col}$ and $\mathbf{S}_{adj}$ respectively, are,*

$$\left(\tilde{\Theta}_{sym}^{(d)}\right)_{ij} = \sqrt{\pi_i\pi_j} \left[ \frac{1 - \left(\sqrt{\pi_i\pi_j}\left(1 + \delta_{ij}r^2\right)\right)^{d+1}}{1 - \sqrt{\pi_i\pi_j}\left(1 + \delta_{ij}r^2\right)} + \delta_{ij}r^{2(d+1)}\frac{1 - \left(\sqrt{\pi_i\pi_j}\left(1 + \delta_{ij}r^2\right)r^{-2}\right)^{d+1}}{1 - \sqrt{\pi_i\pi_j}\left(1 + \delta_{ij}r^2\right)r^{-2}} \right],$$

$$\left(\tilde{\Theta}_{row}^{(d)}\right)_{ij} = 2\gamma \left[ \frac{1 - \left(2\gamma\left(1 + \delta_{ij}r^2\right)\right)^{d+1}}{1 - 2\gamma\left(1 + \delta_{ij}r^2\right)} + \delta_{ij}r^{2(d+1)}\frac{1 - \left(2\gamma\left(1 + \delta_{ij}r^2\right)r^{-2}\right)^{d+1}}{1 - 2\gamma\left(1 + \delta_{ij}r^2\right)r^{-2}} \right],$$

$$\left(\tilde{\Theta}_{col}^{(d)}\right)_{ij} = n\pi_i\pi_j \left[ \frac{1 - \left(n\pi_i\pi_j\left(1 + \delta_{ij}r^2\right)\right)^{d+1}}{1 - n\pi_i\pi_j\left(1 + \delta_{ij}r^2\right)} + \delta_{ij}r^{2(d+1)}\frac{1 - \left(n\pi_i\pi_j\left(1 + \delta_{ij}r^2\right)r^{-2}\right)^{d+1}}{1 - n\pi_i\pi_j\left(1 + \delta_{ij}r^2\right)r^{-2}} \right],$$

$$\left(\tilde{\Theta}_{adj}^{(d)}\right)_{ij} = \pi_i\pi_j \sum_{k=1}^{d+1} \frac{\gamma^{2^k+d-k}}{n^{2k}} \left(\mathbb{1}[\delta_{ij} = 1]\left(p^2 + q^2\right) + \mathbb{1}[\delta_{ij} = -1]\left(2pq\right)\right)^{d+1-k} \times$$
$$\sum_{l=0}^{2^{k-1}} \mathbb{1}[\delta_{ij} = 1]\binom{2^k}{2l}p^{2^k-2l}q^{2l} + \mathbb{1}[\delta_{ij} = -1]\binom{2^k}{2l+1}p^{2^k-2l-1}q^{2l+1}.$$

Note that the two assumptions on $\boldsymbol{\pi}$ are only to express the kernel in a simplified, easy to comprehend format. It is derived without the assumptions on $\boldsymbol{\pi}$ in Appendix A.2.2. Furthermore, the numerical validation of our result is without both these assumptions (Section 3.2).

**Comparison of graph convolutions.** The population NTKs $\tilde{\Theta}^{(d)}$ of depth $d$ in Theorem 2 describes the information that the kernel has after $d$ convolutions with $\mathbf{S}$. To classify the nodes perfectly, the *kernel should ideally have a block structure that aligns with the DC-SBM ($p$ and $q$ blocks)* unaffected by degree correction $\boldsymbol{\pi}$, showing *class separability*, that is, gap between in-class and out-of-class blocks proportional to $p - q$. On this basis, only $\tilde{\Theta}_{row}$ exhibits a block structure unaffected by the degree correction $\boldsymbol{\pi}$, and the gap is determined by $r^2$ and $d$, making $\mathbf{S}_{row}$ preferable over $\mathbf{S}_{sym}$, $\mathbf{S}_{adj}$ and $\mathbf{S}_{col}$. On the other hand, $\tilde{\Theta}_{sym}$, $\tilde{\Theta}_{col}$ and $\tilde{\Theta}_{adj}$ are influenced by the degree correction which obscures the class information especially with depth. Although $\tilde{\Theta}_{sym}$ and $\tilde{\Theta}_{col}$ seem similar, $\tilde{\Theta}_{col}$ is additionally influenced by the number of nodes $n$ in the graph, making it undesirable over $\mathbf{S}_{sym}$. As a result, the preference order from the theory is $\tilde{\Theta}_{row} \succ \tilde{\Theta}_{sym} \succ \tilde{\Theta}_{col} \succ \tilde{\Theta}_{adj}$.

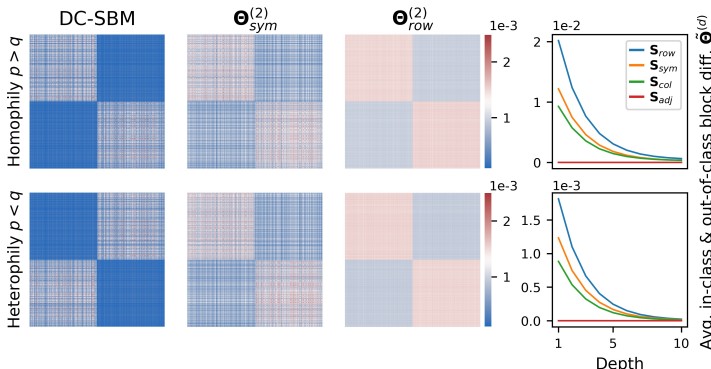

Figure 2: Numerical validation of Theorem 2 using homophilic ($q < p$) and heterophilic ($p < q$) DC-SBM (Column 1). Columns 2 and 3 illustrate the exact NTKs of depth=2 for $\mathbf{S}_{sym}$ and $\mathbf{S}_{row}$, respectively. Column 4 shows the average gap between in-class and out-of-class blocks from theory.

## 3.1 Impact of Depth in Vanilla GCN

Given that $r = \frac{p-q}{p+q} < 1$, Theorem 2 shows that the difference between in-class and out-of-class blocks decreases with depth monotonically which in turn leads to decrease in performance with depth, therefore explaining the observation in Figure 1. The kernel in the limit is stated below.

**Corollary 1 (Population NTK $\tilde{\Theta}^{(\infty)}$ as $d \to \infty$ )** *From Theorem 2,* $\left(\tilde{\Theta}^{(\infty)}_{adj}\right)_{ij} = 0$ *and* $\forall\, i, j$ *for*
$conv \in \{sym, row, col\}$, $\left(\tilde{\Theta}^{(\infty)}_{conv}\right)_{ij} = \dfrac{\nu_{ij}}{1 - \nu_{ij}\left(1 + \delta_{ij}r^2\right)}$ *where* $\nu_{ij} = \sqrt{\pi_i \pi_j}$ *for sym,* $\nu_{ij} = 2\gamma$ *for row and* $\nu_{ij} = n\pi_i \pi_j$ *for col.*

From the corollary, we infer that the class separability at infinite depth is $0$ for $\mathbf{S}_{adj}$, and $\mathcal{O}(r^2)$ for $\mathbf{S}_{sym}$, $\mathbf{S}_{row}$ and $\mathbf{S}_{col}$ showing that the large depth GCN has very little to zero class information. To further understand this, we plot the average in-class and out-of-class block difference for homophilic and heterophilic graphs using the theoretically derived population NTK $\tilde{\Theta}^{(d)}$ for depths $[1, 10]$ in a well separated DC-SBM (Column 4 of Figure 2). It clearly shows the rapid degradation of class separability with depth and the gap goes to $0$ for large depths in all the four convolutions. Additionally, the gap in $\tilde{\Theta}^{(d)}_{row}$ is the highest showing that the class information is better preserved, illustrating the strong representation power of $\mathbf{S}_{row}$. Consequently, *large depth is undesirable for all the four graph convolutions in vanilla GCN and the theory suggests $\mathbf{S}_{row}$ as the best choice for shallow GCN.*

## 3.2 Numerical Validation for Random Graphs

Theorem 2 and Corollary 1 show that $\mathbf{S}_{row}$ has better representation power under Assumption 1 and 2, that is, for the linear GCN with orthonormal features and population DC-SBM. We validate this on homophilous and heterophilous random graphs generated from DC-SBM shown in column 1 of Figure 2. A graph of $n = 1000$ nodes with equal sized classes is sampled from each DC-SBM, respectively. The heatmaps for depth=2 in the case of both homophily and heterophily graphs show that the class information for all the nodes is well preserved in $\mathbf{S}_{row}$ as there is a clear block structure than $\mathbf{S}_{sym}$ in which each node is diffused unequally due to the degree correction. Thus validating the results derived from population NTK. Appendix B.3 presents the results for $\mathbf{S}_{adj}$ and $\mathbf{S}_{col}$ where both are uninformative and behave as derived theoretically.

## 4 Skip Connections Retain Information Even at Infinite Depth

Skip connection is the most common way to overcome the performance degradation with depth in GCNs, but little is known about the effectiveness of different skip connections and their interplay with the convolutions. While our focus is to understand the interplay with convolutions, we also include the impact of convolving with and without the feature information. Hence, we consider the

following two variants: Skip-PC (pre-convolution), where the skip is added to the features before applying convolution (Kipf & Welling, 2017); and Skip-$\alpha$, which gives importance to the features by adding it to each layer without convolving with $\mathbf{S}$ (Chen et al., 2020). To facilitate skip connections, we need to enforce constant layer size, that is, $h_i = h_{i-1}$. Therefore, we transform the input layer using a random matrix $\mathbf{W}$ to $\mathbf{H}_0 = \mathbf{XW}$ of size $n \times h$ where $\mathbf{W}_{ij} \sim \mathcal{N}(0,1)$ and $h$ is the hidden layer size. Let $\mathbf{H}_i$ be the output of layer $i$.

**Definition 2 (Skip-PC)** *In a Skip-PC (pre-convolution) network, the transformed input $\mathbf{H}_0$ is added to the hidden layers before applying the graph convolution $\mathbf{S}$, that is, $\mathbf{H}_i :=$ $\sqrt{\frac{c_\sigma}{h}}\mathbf{S}\left(\mathbf{H}_{i-1} + \sigma_s\left(\mathbf{H}_0\right)\right)\mathbf{W}_i \; \forall i \in [d]$, where $\sigma_s(.)$ can be linear or ReLU.*

The above definition deviates from Kipf & Welling (2017) in the fact that we skip to the input layer instead of the previous layer. The following defines the skip connection similar to Chen et al. (2020).

**Definition 3 (Skip-$\alpha$)** *Given an interpolation coefficient $\alpha \in (0,1)$, a Skip-$\alpha$ network is defined such that the transformed input $\mathbf{H}_0$ and the hidden layer are interpolated linearly, that is, $\mathbf{H}_i :=$ $\sqrt{\frac{c_\sigma}{h}}\left((1-\alpha)\,\mathbf{SH}_{i-1} + \alpha\sigma_s\left(\mathbf{H}_0\right)\right)\mathbf{W}_i \; \forall i \in [d]$, where $\sigma_s(.)$ can be linear or ReLU.*

### 4.1 NTK for GCN with Skip Connections

We derive NTKs for the skip connections – Skip-PC and Skip-$\alpha$ by considering the hidden layers width $h \to \infty$. Both the NTKs maintain the form presented in Theorem 1 with the following changes to the co-variance matrices. Let $\tilde{\mathbf{E}}_0 = \underset{\mathbf{F} \sim \mathcal{N}(\mathbf{0},\boldsymbol{\Sigma}_0)}{\mathbb{E}}\left[\sigma_s(\mathbf{F})\sigma_s(\mathbf{F})^T\right]$.

**Corollary 2 (NTK for Skip-PC)** *The NTK for an infinitely wide Skip-PC network is as presented in Theorem 1 where $\mathbf{E}_k$ is defined as in the theorem, but $\boldsymbol{\Sigma}_k$ is defined as*

$$\boldsymbol{\Sigma}_0 = \mathbf{XX}^T, \qquad \boldsymbol{\Sigma}_1 = \mathbf{S}\tilde{\mathbf{E}}_0\mathbf{S}^T \qquad and \qquad \boldsymbol{\Sigma}_k = \mathbf{SE}_{k-1}\mathbf{S}^T + \boldsymbol{\Sigma}_1.$$

**Corollary 3 (NTK for Skip-$\alpha$)** *The NTK for an infinitely wide Skip-$\alpha$ network is as presented in Theorem 1 where $\mathbf{E}_k$ is defined as in the theorem, but $\boldsymbol{\Sigma}_k$ is defined with $\boldsymbol{\Sigma}_0 = \mathbf{XX}^T$,*

$$\boldsymbol{\Sigma}_1 = (1-\alpha)^2\,\mathbf{SE}_0\mathbf{S}^T + \alpha\,(1-\alpha)\left(\mathbf{SE}_0 + \mathbf{E}_0\mathbf{S}^T\right) + \alpha^2\mathbf{E}_0 \; and \; \boldsymbol{\Sigma}_k = (1-\alpha)^2\mathbf{SE}_{k-1}\mathbf{S}^T + \alpha^2\tilde{\mathbf{E}}_0.$$

### 4.2 Impact of Depth in GCNs with Skip Connection

Similar to the previous section we use the NTK for Skip-PC and Skip-$\alpha$ (Corollary 2 and 3) and analyze the graph convolutions $\mathbf{S}_{sym}$ and $\mathbf{S}_{adj}$ under the same considerations detailed in Section 3. Since, $\mathbf{S}_{adj}$ and $\mathbf{S}_{col}$ are theoretically worse and not popular in practice, we do not consider them for the skip connection analysis. The linear orthonormal feature NTK, $\boldsymbol{\Theta}^{(d)}$, for depth $d$ is same as (4) with changes to $\boldsymbol{\Sigma}_k$ as follows,

Skip-PC: $\boldsymbol{\Sigma}_k = \mathbf{S}^k\mathbf{S}^{kT} + \mathbf{SS}^T$,

Skip-$\alpha$: $\boldsymbol{\Sigma}_k = (1-\alpha)^{2k}\,\mathbf{S}^k\mathbf{S}^{kT} + \alpha\,(1-\alpha)^{2k-1}\,\mathbf{S}^{k-1}\left(\mathbf{S} + \mathbf{S}^T\right)\mathbf{S}^{k-1^T} + \alpha^2\sum_{l=0}^{k-1}(1-\alpha)^{2l}\,\mathbf{S}^l\mathbf{S}^{lT}$.

We derive the population NTK $\tilde{\boldsymbol{\Theta}}^{(d)}$ and, for convenience, only state the result as $d \to \infty$ in the following theorems.

**Theorem 3 (Population NTK for Skip-PC $\tilde{\boldsymbol{\Theta}}_{PC}^{(\infty)}$)** *Under the assumptions of Theorem 2,*

$$\left(\tilde{\boldsymbol{\Theta}}_{PC,sym}^{(\infty)}\right)_{ij} = \frac{\sqrt{\pi_i\pi_j}\left(2 + \delta_{ij}r^2\right)}{1 - \sqrt{\pi_i\pi_j}\left(1 + \delta_{ij}r^2\right)}, \quad and \quad \left(\tilde{\boldsymbol{\Theta}}_{PC,row}^{(\infty)}\right)_{ij} = \frac{2\gamma(2 + \delta_{ij}r^2)}{1 - 2\gamma\left(1 + \delta_{ij}r^2\right)}. \tag{5}$$

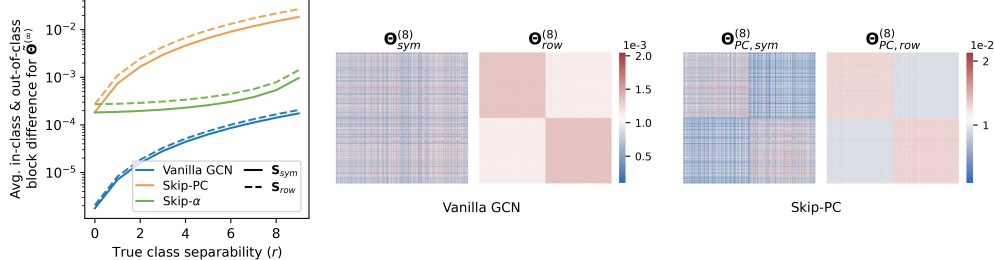

Figure 3: Left: average in-class and out-of-class block difference at $d = \infty$ (in log scale) for different true class separability. Heatmaps: $\Theta^{(8)}$ for $\mathbf{S}_{sym}$ and $\mathbf{S}_{row}$ for vanilla GCN and Skip-PC.

**Theorem 4 (Population NTK for Skip-$\alpha$ $\tilde{\Theta}_{\alpha}^{(\infty)}$ )** *Under the assumptions of Theorem 2,*

$$\left(\tilde{\Theta}_{\alpha,sym}^{(\infty)}\right)_{ij} = \frac{\alpha^2\sqrt{\pi_i\pi_j}}{1 - \sqrt{\pi_i\pi_j}\left(1 + \delta_{ij}r^2\right)}\left(\frac{1}{1 - (1-\alpha)^2} + \frac{\delta_{ij}}{1 - (1-\alpha)^2 r^2}\right), \quad and$$

$$\left(\tilde{\Theta}_{\alpha,row}^{(\infty)}\right)_{ij} = \frac{2\gamma\alpha^2}{1 - 2\gamma\left(1 + \delta_{ij}r^2\right)}\left(\frac{1}{1 - (1-\alpha)^2} + \frac{\delta_{ij}}{1 - (1-\alpha)^2 r^2}\right). \tag{6}$$

Similar to Theorem 2, assumptions on $\boldsymbol{\pi}$ in above theorems is to simplify the results. To understand the role of skip connections, we plot the gap between in-class and out-of-class blocks at infinite depth for different values of true class separability $r$, for vanilla GCN, Skip-PC and Skip-$\alpha$ using Corollary 1, Theorems 3–4, respectively (Figure 3). The plot clearly shows that the gap is away from 0 for both the skip connections given a reasonable true separation, unlike vanilla GCN. This implies the class information is retained in skip connections even at infinite depth.

### 4.3 Numerical Validation for Random Graphs

We validate our theoretical result using the same setup detailed in Section 3.2 without the assumptions, and compute the exact NTKs for Skip-PC and Skip-$\alpha$ for both $\mathbf{S}_{sym}$ and $\mathbf{S}_{row}$. We show the result on homophilic graphs but they equally extend to the heterophilic case. While $\mathbf{S}_{sym}$ has no class information for depth=8 in vanilla GCN (Figure 3 middle), it is retained well in Skip-PC (right plot). In the case of $\mathbf{S}_{row}$, we clearly observe the blocks in both cases with more prevalent gap in Skip-PC illustrating our theoretical results. Similar observation is made for Skip-$\alpha$ despite considering $\mathbf{XX}^T = \mathbf{I}_n$ as the model interpolates with the feature, and is discussed in Appendix B.3. While both $\mathbf{S}_{sym}$ and $\mathbf{S}_{row}$ retain the class information in larger depths, we observe that the degree correction plays a significant role in $\mathbf{S}_{sym}$ as elucidated in our theoretical analysis.

## 5 $\mathbf{S}_{sym}$ Maybe Preferred Over $\mathbf{S}_{row}$ in Absence of Class Structure

While we showed that the graph convolution $\mathbf{S}_{row}$ preserves the underlying class structure, it is natural to wonder about the random graphs that have no communities ($p = q$). One such case is graphs with core-periphery structure where the graph has core nodes that are highly interconnected and periphery nodes that are sparsely connected to the core and other periphery nodes. Such a graph can be modeled using only the degree correction $\boldsymbol{\pi}$ such that $\pi_j \ll \pi_i \, \forall j \in periphery, i \in core$ (similar to Jia & Benson (2019)). Extending Theorem 2, we derive the following Corollary 4 and show that the convolution $\mathbf{S}_{sym}$ contains the graph information while $\mathbf{S}_{row}$ is a constant kernel.

**Corollary 4 (Population NTKs $\tilde{\Theta}$ for $p = q$)** *Let Assumption 1 and 2 hold, $K = 2$ and $p = q$. Furthermore, $\boldsymbol{\pi}$ is chosen such that $\sum_{i \in core} \pi_i^2 = \lambda$ and $\sum_{i \in periphery} \pi_i^2 = \mu$. Then $\forall i$ and $j$, the population NTKs $\tilde{\Theta}_{sym}$ and $\tilde{\Theta}_{row}$ of depth $d$ for $\mathbf{S} = \mathbf{S}_{sym}$ and $\mathbf{S}_{row}$, respectively, are,*

$$\left(\tilde{\Theta}_{sym}^{(d)}\right)_{ij} = \sqrt{\pi_i\pi_j}\frac{1 - \left(\sqrt{\pi_i\pi_j}\right)^{d+1}}{1 - \sqrt{\pi_i\pi_j}} \quad and \quad \left(\tilde{\Theta}_{row}^{(d)}\right)_{ij} = (\lambda + \mu)\frac{1 - (\lambda + \mu)^{d+1}}{1 - (\lambda + \mu)}.$$

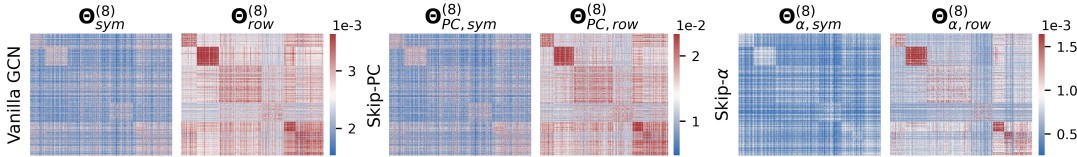

Figure 4: Evaluation on Cora dataset. Heatmaps show results of vanilla GCN, Skip-PC and Skip-$\alpha$ where a min and max threshold of 10 and 90 percentile is set for better visualization.

From Corollary 4, it is evident that the $\mathbf{S}_{sym}$ has the graph information and hence could be preferred when there is no community structure. Furthermore, interestingly even skip connections prove to be of no use for $\mathbf{S}_{row}$ and it remains a constant kernel in this case as well. We validate it experimentally and discuss the results in Appendix B.3 (Figure 10). While $\mathbf{S}_{row}$ results in a constant kernel for core-periphery without community structure, it is important to note that when there exists a community structure and each community has core-periphery nodes, then $\mathbf{S}_{row}$ is still preferable over $\mathbf{S}_{sym}$ as it is simply a special case of homophilic networks. This is demonstrated in Appendix B.3 (Figure 11).

## 6 EMPIRICAL ANALYSIS ON REAL DATA

In this section, we explore how well the theoretical results translate to real dataset *Cora* with features, that is, $\mathbf{X}\mathbf{X}^T \neq \mathbf{I}_n$ and $\mathbf{A} \neq \mathbf{M}$. We also provide experimental details and additional experiments on *Citeseer* in Appendix B.5. We consider multi-class node classification for Cora ($K = 7$) using GCN with linear activations and provide the results for ReLU activations in Appendix B.4 The NTKs for vanilla GCN, GCN with Skip-PC and Skip-$\alpha$ are illustrated in Figure 4. We make the following observations from the experiments that validate the theory even in a much relaxed setting, (i) clear block structures show up in both GCN with and without skip connections for $\mathbf{S}_{row}$, thus illustrating that the class information is well retained by $\mathbf{S}_{row}$ than $\mathbf{S}_{sym}$; (ii) while we cannot compare the skip connections, it is still evident that $\mathbf{S}_{row}$ is better than $\mathbf{S}_{sym}$ for both Skip-PC and Skip-$\alpha$ as block structures emerge even in the case of large depth. Thus, although the theoretical result is based on DC-SBM with mild assumptions, the conclusions hold well in real settings on real datasets as well.

## 7 CONCLUSION

Graph convolution operators significantly influence the performance of GCNs, but existing learning theoretic bounds for GCNs do not provide insight into the representation power of the operators. We present a NTK based analysis that characterizes different convolutions, thereby proving the strong representation power of $\mathbf{S}_{row}$ in community detection and explaining why $\mathbf{S}_{row}$, and to some extent $\mathbf{S}_{sym}$, are preferred in practice (Theorem 2). In contrast to applying spectral analysis of the convolutions to explain over-smoothing, our explicit characterization of the network provides more exact quantification of the impact of over-smoothing in deep GCNs (Corollary 1, see Figure 2). In addition, the NTKs for GCNs with skip connections enable precise understanding of the role of skip connections in countering the over-smoothing effect (Theorems 3–4). While the DC-SBM assumption may seem restrictive, experiments on Cora and Citeseer show that our theoretical results hold beyond DC-SBM, although formally characterizing such behavior could be difficult without model assumptions. We note that our analysis could be extended by considering feature information ($\mathbf{X}\mathbf{X}^T \neq \mathbf{I}_n$) or random samples from DC-SBM, which would require more involved analysis but could provide further insights into GCNs, such as interplay between graph and feature information.

The present NTK based setup allows for the analysis of different graphs having homophilic, heterophilic and core-periphery structures, and can be extended to other graph generating processes. Furthermore, the general formulation of NTK for vanilla GCNs (Theorem 1) and with skip connections (Corollaries 2–3) can be used for analyzing any new convolutions like topological structure preserving convolutions, for obtaining a rigorous understanding of GCNs by deriving statistical consistency results or information theoretic limits, as well as for theoretical analysis of other graph learning problems, such as link prediction.

## 8 ETHICS STATEMENT

Our work focuses on understanding some characteristics of the graph neural network theoretically and hence it doesn't have direct implications on the ethical and fairness aspects.

## 9 REPRODUCIBILITY STATEMENT

The assumptions for the theory are stated clearly in Assumptions 1–2, and all the theoretical results, Theorems 1–4, Corollaries 1–3, are proved in detail in Appendix A. The implementation of GCN and NTK for GCNs with and without skip connections are provided in `ntk_gcn_conv.zip` as a supplementary material. Datasets used in the experiments are publicly available and also provided in `data` folder available in the zip. The experimental results can be reproduced by following the instructions in `readme.md`.

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

## A    Mathematical derivations and proofs

We first derive the NTK (Theorem 1) for GCN defined in (2) and prove Theorem 2 by considering linear GCN and computing the population NTK $\tilde{\boldsymbol{\Theta}}^{(d)}$ for different graph convolutions. We use $\mathbf{1}_n$ to denote a vector of $n$ dimension with all 1s and $\hat{\mathbf{1}}_n$ for a vector of $n$ dimension with $-1$ as first $\frac{n}{2}$ entries and $+1$ as the remaining $\frac{n}{2}$ entries.

### A.1    Theorem 1: NTK for Vanilla GCN

We rewrite the GCN $F_{\mathbf{W}}(\mathbf{X}, \mathbf{S})$ defined in (2) using the following recursive definitions:

$$\mathbf{G}_1 = \mathbf{S}\mathbf{X}, \qquad \mathbf{G}_i = \sqrt{\frac{c_\sigma}{h_{i-1}}} \mathbf{S}\sigma(\mathbf{F}_{i-1}) \; \forall i \in \{2, \ldots, d+1\}, \quad \mathbf{F}_i = \mathbf{G}_i \mathbf{W}_i \; \forall i \in [d+1]. \quad (7)$$

Thus, $F_{\mathbf{W}}(\mathbf{X}, \mathbf{S}) = \mathbf{F}_{d+1}$ and using the definitions in (7), the gradient with respect to $\mathbf{W}_i$ is

$$\frac{\partial F_{\mathbf{W}}(\mathbf{X}, \mathbf{S})}{\partial \mathbf{W}_i} = \mathbf{G}_i^T \mathbf{B}_i \quad \text{with} \quad \mathbf{B}_{d+1} = \mathbf{1}_n, \qquad \mathbf{B}_i = \sqrt{\frac{c_\sigma}{h_i}} \mathbf{S}^T \mathbf{B}_{i+1} \mathbf{W}_{i+1}^T \odot \dot{\sigma}(\mathbf{F}_i). \quad (8)$$

We derive the NTK, as defined in (1), using the recursive definition of $F_{\mathbf{W}}(\mathbf{X}, \mathbf{S})$ in (7) and its derivative in (8).

**Co-variance between Nodes.** We will first derive the co-variance matrix of size $n \times n$ for each layer comprising of co-variance between any two nodes $u$ and $v$. The co-variance between $u$ and $v$ in $\mathbf{F}_1$ and $\mathbf{F}_i$ are derived below. We denote $u$-th row of matrix $\mathbf{Z}$ as $\mathbf{Z}_{u.}$ throughout our proofs.

$$\mathbb{E}\left[(\mathbf{F}_1)_{uk} (\mathbf{F}_1)_{vk'}\right] = \mathbb{E}\left[(\mathbf{G}_1 \mathbf{W}_1)_{uk} (\mathbf{G}_1 \mathbf{W}_1)_{vk'}\right]$$

$$= \mathbb{E}\left[\sum_{r=1}^{h_0} (\mathbf{G}_1)_{ur} (\mathbf{W}_1)_{rk} \sum_{s=1}^{h_0} (\mathbf{G}_1)_{vs} (\mathbf{W}_1)_{sk'}\right] \overset{(\mathbf{W}_1)_{xy} \sim \mathcal{N}(0,1)}{=} 0 \quad ; \text{if } r \neq s \text{ or } k \neq k'$$

$$\mathbb{E}\left[(\mathbf{F}_1)_{uk} (\mathbf{F}_1)_{vk}\right] \overset{r \equiv s}{\underset{k = k'}{=}} \mathbb{E}\left[\sum_{r=1}^{h_0} (\mathbf{G}_1)_{ur} (\mathbf{G}_1)_{vr} (\mathbf{W}_1)_{rk}^2\right]$$

$$\overset{(\mathbf{W}_1)_{xy} \sim \mathcal{N}(0,1)}{=} \sum_{r=1}^{h_0} (\mathbf{G}_1)_{ur} (\mathbf{G}_1)_{vr} = \langle (\mathbf{G}_1)_{u.}, (\mathbf{G}_1)_{v.} \rangle \quad (9)$$

$$\mathbb{E}\left[(\mathbf{F}_i)_{uk} (\mathbf{F}_i)_{vk}\right] \overset{r \equiv s}{\underset{k = k'}{=}} \mathbb{E}\left[\sum_{r=1}^{h_{i-1}} (\mathbf{G}_i)_{ur} (\mathbf{G}_i)_{vr} (\mathbf{W}_i)_{rk}^2\right]$$

$$\overset{(\mathbf{W}_i)_{xy} \sim \mathcal{N}(0,1)}{=} \sum_{r=1}^{h_{i-1}} (\mathbf{G}_i)_{ur} (\mathbf{G}_i)_{vr} = \langle (\mathbf{G}_i)_{u.}, (\mathbf{G}_i)_{v.} \rangle \quad (10)$$

Evaluating (9) and (10) in terms of the graph in the following,

$$(9): \quad \langle (\mathbf{G}_1)_{u.}, (\mathbf{G}_1)_{v.} \rangle = \langle (\mathbf{S}\mathbf{X})_{u.}, (\mathbf{S}\mathbf{X})_{v.} \rangle = \mathbf{S}_{u.} \mathbf{X}\mathbf{X}^T \mathbf{S}_{.v}^T = (\mathbf{\Sigma}_1)_{uv} \quad (11)$$

$$(10): \quad \langle (\mathbf{G}_i)_{u.}, (\mathbf{G}_i)_{v.} \rangle = \frac{c_\sigma}{h_{i-1}} \langle (\mathbf{S}\sigma(\mathbf{F}_{i-1}))_{u.}, (\mathbf{S}\sigma(\mathbf{F}_{i-1}))_{v.} \rangle$$

$$= \frac{c_\sigma}{h_{i-1}} \sum_{k=1}^{h_{i-1}} (\mathbf{S}\sigma(\mathbf{F}_{i-1}))_{uk} (\mathbf{S}\sigma(\mathbf{F}_{i-1}))_{vk}$$

$$\overset{h_{i-1} \to \infty}{=} c_\sigma \mathbb{E}\left[(\mathbf{S}\sigma(\mathbf{F}_{i-1}))_{uk} (\mathbf{S}\sigma(\mathbf{F}_{i-1}))_{vk}\right] \quad ; \text{law of large numbers}$$

$$= c_\sigma \mathbb{E}\left[\left(\sum_{r=1}^{n} \mathbf{S}_{ur} \sigma(\mathbf{F}_{i-1})_{rk}\right)\left(\sum_{s=1}^{n} \mathbf{S}_{vs} \sigma(\mathbf{F}_{i-1})_{sk}\right)\right]$$

$$= c_\sigma \mathbb{E}\left[\sum_{r=1}^{n} \sum_{s=1}^{n} \mathbf{S}_{ur} \mathbf{S}_{vs} \sigma(\mathbf{F}_{i-1})_{rk} \sigma(\mathbf{F}_{i-1})_{sk}\right]$$

$$\stackrel{(a)}{=} \sum_{r=1}^{n} \sum_{s=1}^{n} \mathbf{S}_{ur} \left(\mathbf{E}_{i-1}\right)_{rs} \mathbf{S}_{sv}^{T} = \mathbf{S}_{u.} \mathbf{E}_{i-1} \mathbf{S}_{.v}^{T} = \left(\boldsymbol{\Sigma}_{i}\right)_{uv} \tag{12}$$

$(a)$: using $\mathbb{E}\left[\left(\mathbf{F}_{i-1}\right)_{rk} \left(\mathbf{F}_{i-1}\right)_{sk}\right] = \left(\boldsymbol{\Sigma}_{i-1}\right)_{rs}$ and the definition of $\mathbf{E}_{i-1}$ in Theorem 1.

**NTK for Vanilla GCN.** Let us first evaluate the tangent kernel component from $\mathbf{W}_i$ respective to nodes $u$ and $v$. The following two results are needed to derive it.

**Result 1 (Inner Product of Matrices).** Let $\mathbf{a}$ and $\mathbf{b}$ be vectors of size $d_1 \times 1$ and $d_2 \times 1$, then

$$\langle \mathbf{ab}^{T}, \mathbf{ab}^{T} \rangle = \mathrm{tr} \left(\mathbf{ab}^{T} \left(\mathbf{ab}^{T}\right)^{T}\right)$$
$$= \mathrm{tr} \left(\mathbf{ab}^{T} \mathbf{ba}^{T}\right) = \mathrm{tr} \left(\mathbf{a}^{T} \mathbf{ab}^{T} \mathbf{b}\right) = \left(\mathbf{a}^{T} \mathbf{a}\right) \odot \left(\mathbf{b}^{T} \mathbf{b}\right) = \langle \mathbf{a}, \mathbf{a} \rangle \odot \langle \mathbf{b}, \mathbf{b} \rangle \tag{13}$$

**Result 2 $\langle \left(B_r\right)_{u.}, \left(B_r\right)_{v.} \rangle$.** We evaluate $\langle \left(\mathbf{B}_r\right)_{u.}, \left(\mathbf{B}_r\right)_{v.} \rangle = \left(\mathbf{B}_r \mathbf{B}_r^{T}\right)_{uv}$ appearing in the gradient.

$$\left(\mathbf{B}_r \mathbf{B}_r^{T}\right)_{uv} = \frac{c_\sigma}{h_r} \sum_{k=1}^{h_r} \left(\mathbf{S}^{T} \mathbf{B}_{r+1} \mathbf{W}_{r+1}^{T}\right)_{uk} \dot{\sigma}(\mathbf{F}_r)_{uk} \left(\mathbf{S}^{T} \mathbf{B}_{r+1} \mathbf{W}_{r+1}^{T}\right)_{vk} \dot{\sigma}(\mathbf{F}_r)_{vk}$$

$$= \frac{c_\sigma}{h_r} \sum_{k=1}^{h_r} \sum_{i,j}^{n,h_{r+1}} \mathbf{S}_{iu} \left(\mathbf{B}_{r+1}\right)_{ij} \left(\mathbf{W}_{r+1}\right)_{kj} \dot{\sigma}(\mathbf{F}_r)_{uk} \dot{\sigma}(\mathbf{F}_r)_{vk} \sum_{i',j'}^{n,h_{r+1}} \mathbf{S}_{i'v} \left(\mathbf{B}_{r+1}\right)_{i'j'} \left(\mathbf{W}_{r+1}\right)_{kj'}$$

$$= \frac{c_\sigma}{h_r} \sum_{i,j}^{n,h_{r+1}} \sum_{i',j'}^{n,h_{r+1}} \left(\mathbf{B}_{r+1}\right)_{ij} \left(\mathbf{B}_{r+1}\right)_{i'j'} \mathbf{S}_{iu} \mathbf{S}_{i'v} \sum_{k=1}^{h_r} \left(\mathbf{W}_{r+1}\right)_{kj} \dot{\sigma}(\mathbf{F}_r)_{uk} \dot{\sigma}(\mathbf{F}_r)_{vk} \left(\mathbf{W}_{r+1}\right)_{kj'}$$

$$= \sum_{j,j'}^{h_{r+1},h_{r+1}} \left(\mathbf{S}^{T} \mathbf{B}_{r+1}\right)_{uj} \left(\mathbf{S}^{T} \mathbf{B}_{r+1}\right)_{vj'} \frac{c_\sigma}{h_r} \sum_{k=1}^{h_r} \left(\mathbf{W}_{r+1}\right)_{kj} \dot{\sigma}(\mathbf{F}_r)_{uk} \dot{\sigma}(\mathbf{F}_r)_{vk} \left(\mathbf{W}_{r+1}\right)_{kj'}$$

$$\stackrel{h_r \to \infty}{=} \sum_{j}^{h_{r+1}} \left(\mathbf{S}^{T} \mathbf{B}_{r+1}\right)_{uj} \left(\mathbf{S}^{T} \mathbf{B}_{r+1}\right)_{vj} c_\sigma \mathbb{E}\left[\left(\mathbf{W}_{r+1}^2\right)_{kj} \dot{\sigma}(\mathbf{F}_r)_{uk} \dot{\sigma}(\mathbf{F}_r)_{vk}\right] \quad ; 0 \text{ for } j \neq j'$$

$$\stackrel{(b)}{=} \langle \left(\mathbf{S}^{T} \mathbf{B}_{r+1}\right)_{u.}, \left(\mathbf{S}^{T} \mathbf{B}_{r+1}\right)_{v.} \rangle c_\sigma \mathbb{E}\left[\dot{\sigma}(\mathbf{F}_r)_{uk} \dot{\sigma}(\mathbf{F}_r)_{vk}\right]$$

$$\stackrel{(13)}{=} \left(\mathbf{SS}^{T}\right)_{uv} \langle \mathbf{B}_{r+1}, \mathbf{B}_{r+1} \rangle_{uv} c_\sigma \mathbb{E}\left[\dot{\sigma}(\mathbf{F}_r)_{uk} \dot{\sigma}(\mathbf{F}_r)_{vk}\right]$$

$$= \left(\mathbf{SS}^{T}\right)_{uv} \langle \mathbf{B}_{r+1}, \mathbf{B}_{r+1} \rangle_{uv} \left(\dot{\mathbf{E}}_r\right)_{uv} \tag{14}$$

$(b)$: $\left(\mathbf{W}_{r+1}\right)_{kj}$ is independent and $\mathbb{E}\left[\left(\mathbf{W}_{r+1}^2\right)_{kj} = 1\right]$.

Now, lets derive $\left\langle \left(\frac{\partial \mathbf{F}}{\partial \mathbf{W}_k}\right)_u, \left(\frac{\partial \mathbf{F}}{\partial \mathbf{W}_k}\right)_v \right\rangle$ and $\left\langle \left(\frac{\partial \mathbf{F}}{\partial \mathbf{W}_1}\right)_u, \left(\frac{\partial \mathbf{F}}{\partial \mathbf{W}_1}\right)_v \right\rangle$ using the above results.

$$\left\langle \left(\frac{\partial \mathbf{F}}{\partial \mathbf{W}_k}\right)_u, \left(\frac{\partial \mathbf{F}}{\partial \mathbf{W}_k}\right)_v \right\rangle = \left\langle \left(\mathbf{G}_k\right)_{u.}^{T} \left(\mathbf{B}_k\right)_{u.}, \left(\mathbf{G}_k\right)_{v.}^{T} \left(\mathbf{B}_k\right)_{v.} \right\rangle$$

$$\stackrel{(13)}{=} \langle \left(\mathbf{G}_k\right)_{u.}, \left(\mathbf{G}_k\right)_{v.} \rangle \odot \langle \left(\mathbf{B}_k\right)_{u.}, \left(\mathbf{B}_k\right)_{v.} \rangle$$

$$\stackrel{(12),(14)}{=} \left(\boldsymbol{\Sigma}_k\right)_{uv} \left(\mathbf{SS}^{T}\right)_{uv} \langle \mathbf{B}_{r+1}, \mathbf{B}_{r+1} \rangle_{uv} \left(\dot{\mathbf{E}}_r\right)_{uv}$$

$$\stackrel{(c)}{=} \left(\boldsymbol{\Sigma}_k\right)_{uv} \left(\left(\mathbf{SS}^{T}\right)_{uv}\right)^{d+1-k} \left(\prod_{k'=k}^{d+1-k} \left(\dot{\mathbf{E}}_{k'}\right)_{uv}\right) \langle \mathbf{B}_{d+1}, \mathbf{B}_{d+1} \rangle_{uv}$$

$$\stackrel{(d)}{=} \left(\boldsymbol{\Sigma}_k\right)_{uv} \left(\left(\mathbf{SS}^{T}\right)_{uv}\right)^{d+1-k} \left(\prod_{k'=k}^{d+1-k} \left(\dot{\mathbf{E}}_{k'}\right)_{uv}\right) \tag{15}$$

$(c)$: repeated application of (14).
$(d)$: definition of $\mathbf{B}_{d+1}$.

Extending (15) to all $n$ nodes which will result in $n \times n$ matrix,

$$\left\langle \frac{\partial \mathbf{F}}{\partial \mathbf{W}_k}, \frac{\partial \mathbf{F}}{\partial \mathbf{W}_k} \right\rangle = \boldsymbol{\Sigma}_k \odot \left(\mathbf{S}\mathbf{S}^T\right)^{\odot d+1-k} \overset{d+1-k}{\underset{k'=k}{\bigodot}} \dot{\mathbf{E}}_{k'}$$

$$\underset{\mathbf{W}_k}{\mathbb{E}} \left[ \left\langle \frac{\partial \mathbf{F}}{\partial \mathbf{W}_k}, \frac{\partial \mathbf{F}}{\partial \mathbf{W}_k} \right\rangle \right] = \boldsymbol{\Sigma}_k \odot \left(\mathbf{S}\mathbf{S}^T\right)^{\odot d+1-k} \overset{d+1-k}{\underset{k'=k}{\bigodot}} \dot{\mathbf{E}}_{k'} \qquad (16)$$

Finally, NTK $\boldsymbol{\Theta}$ is,

$$\boldsymbol{\Theta} = \sum_{k=1}^{d+1} \underset{\mathbf{W}_k}{\mathbb{E}} \left[ \left\langle \frac{\partial \mathbf{F}}{\partial \mathbf{W}_k}, \frac{\partial \mathbf{F}}{\partial \mathbf{W}_k} \right\rangle \right]$$

$$= \sum_{k=1}^{d+1} \boldsymbol{\Sigma}_k \odot \left(\mathbf{S}\mathbf{S}^T\right)^{\odot(d+1-k)} \odot \left( \overset{d+1-k}{\underset{k'=k}{\bigodot}} \dot{\mathbf{E}}'_k \right) \qquad (17)$$

with definition of $\boldsymbol{\Sigma}_k$ and $\dot{\mathbf{E}}_k$ mentioned in the theorem. $\qquad\qquad\square$

## A.2 THEOREM 2 AND COROLLARY 1: POPULATION NTK $\tilde{\boldsymbol{\Theta}}$ FOR DIFFERENT S

We consider Assumption 1, that is, linear GCN with orthonormal features and Assumption 2 without assumption on $\gamma$. We first prove it for $K = 2$ and then extend it to $K$ classes. We consider that all nodes are sorted per class for ease of analysis which implies $\mathbf{A}$ is a $n \times n$ matrix with $p\pi_i\pi_j$ entries in $[1, \frac{n}{2}][1, \frac{n}{2}]$ and $[\frac{n}{2}+1, n][\frac{n}{2}+1, n]$ blocks and $q\pi_i\pi_j$ entries in $[1, \frac{n}{2}][\frac{n}{2}+1, n]$ and $[\frac{n}{2}+1, n][1, \frac{n}{2}]$ blocks. Therefore,

$$\mathbf{A} = \boldsymbol{\pi}\boldsymbol{\pi}^T \odot \left( \frac{p+q}{2}\mathbf{1}\mathbf{1}^T + \frac{p-q}{2}\hat{\mathbf{1}}\hat{\mathbf{1}}^T \right)$$

$$= \frac{p+q}{2}\boldsymbol{\pi}\boldsymbol{\pi}^T + \frac{p-q}{2}\hat{\boldsymbol{\pi}}\hat{\boldsymbol{\pi}}^T \qquad (18)$$

where the entries of $\hat{\boldsymbol{\pi}}$ are $-\pi_i \,\forall\, i \in [1, \frac{n}{2}]$ and $+\pi_i \,\forall\, i \in [\frac{n}{2}+1, n]$. $\mathbf{D}$ be the degree matrix of $\mathbf{A}$ and $\mathbf{D} = \frac{p+q}{2}\mathrm{diag}(\boldsymbol{\pi})$.

### A.2.1 SYMMETRIC DEGREE NORMALIZED ADJACENCY $\mathbf{S}_{sym}$

Now, lets compute $\mathbf{S}_{sym}$ using $\mathbf{A}$ (18) and its degree matrix $\mathbf{D}$.

$$\mathbf{S}_{sym} = \mathbf{D}^{-\frac{1}{2}}\mathbf{A}\mathbf{D}^{-\frac{1}{2}}$$

$$= \frac{2}{p+q}\mathrm{diag}(\boldsymbol{\pi})^{-\frac{1}{2}} \left( \frac{p+q}{2}\boldsymbol{\pi}\boldsymbol{\pi}^T + \frac{p-q}{2}\hat{\boldsymbol{\pi}}\hat{\boldsymbol{\pi}}^T \right) \mathrm{diag}(\boldsymbol{\pi})^{-\frac{1}{2}}$$

$$= \boldsymbol{\pi}^{\frac{1}{2}}\boldsymbol{\pi}^{\frac{1}{2}T} + \frac{p-q}{p+q}\hat{\boldsymbol{\pi}}^{\frac{1}{2}}\hat{\boldsymbol{\pi}}^{\frac{1}{2}T}$$

$$= \begin{bmatrix} \sqrt{\pi_1} & -\sqrt{\pi_1} \\ \vdots & \vdots \\ \sqrt{\pi_n} & +\sqrt{\pi_n} \end{bmatrix}_{n\times 2} \begin{bmatrix} 1 & 0 \\ 0 & r \end{bmatrix}_{2\times 2} \begin{bmatrix} \sqrt{\pi_1} & -\sqrt{\pi_1} \\ \vdots & \vdots \\ \sqrt{\pi_n} & +\sqrt{\pi_n} \end{bmatrix}^T_{2\times n} \quad ; r = \frac{p-q}{p+q}$$

$$= \mathbf{U}\boldsymbol{\Lambda}\mathbf{U}^T \qquad (19)$$

Note that $\boldsymbol{\pi}^T\boldsymbol{\pi} = \hat{\boldsymbol{\pi}}^T\hat{\boldsymbol{\pi}} = 1$, $\boldsymbol{\pi}^T\hat{\boldsymbol{\pi}} = 0$ and $\mathbf{U}^T\mathbf{U} = \mathbf{I}_2$, thus (19) is the singular value decomposition of $\mathbf{S}_{sym}$.

To compute the population NTK $\tilde{\mathbf{\Theta}}_{sym}^{(d)}$ in (4), we need $\mathbf{S}_{sym}^k \mathbf{S}_{sym}^{kT}$. Using (19),

$$\mathbf{S}_{sym}^k \mathbf{S}_{sym}^{kT} \overset{(19)}{=} \mathbf{U}\mathbf{\Lambda}^{2k}\mathbf{U}^T$$

$$= \begin{bmatrix} \sqrt{\pi_1} & -\sqrt{\pi_1} \\ \vdots & \vdots \\ \sqrt{\pi_n} & +\sqrt{\pi_n} \end{bmatrix}_{n\times 2} \begin{bmatrix} 1 & 0 \\ 0 & r^{2k} \end{bmatrix}_{2\times 2} \begin{bmatrix} \sqrt{\pi_1} & -\sqrt{\pi_1} \\ \vdots & \vdots \\ \sqrt{\pi_n} & +\sqrt{\pi_n} \end{bmatrix}^T_{2\times n}$$

$$\left(\mathbf{S}_{sym}^k \mathbf{S}_{sym}^{kT}\right)_{ij} = \left(1 + \delta_{ij}r^{2k}\right)\sqrt{\pi_i\pi_j} \qquad ; \delta_{ij} = (-1)^{\mathbb{1}[\mathcal{C}_i \neq \mathcal{C}_j]}$$

$$\mathbf{S}_{sym}^k \mathbf{S}_{sym}^{kT} \overset{\text{matrix}}{\underset{\text{notation}}{=}} \left[ \begin{array}{c|c} \left(1+r^{2k}\right)\sqrt{\pi_i\pi_j} & \left(1-r^{2k}\right)\sqrt{\pi_i\pi_j} \\ \hline \left(1-r^{2k}\right)\sqrt{\pi_i\pi_j} & \left(1+r^{2k}\right)\sqrt{\pi_i\pi_j} \end{array} \right]_{n\times n} \tag{20}$$

$$\underbrace{\phantom{xxxxxx}}_{\frac{n}{2}\text{ entries}} \underbrace{\phantom{xxxxxx}}_{\frac{n}{2}\text{ entries}}$$

Consequently, population NTK $\tilde{\mathbf{\Theta}}_{sym}^{(d)}$ for nodes $i$ and $j$ using (20) is as follows,

$$\left(\tilde{\mathbf{\Theta}}_{sym}^{(d)}\right)_{ij} = \sum_{k=1}^{d+1} \sqrt{\pi_i\pi_j}\left(1+\delta_{ij}r^{2k}\right)\left(\sqrt{\pi_i\pi_j}\left(1+\delta_{ij}r^2\right)\right)^{d+1-k}$$

$$= \sum_{k=1}^{d+1} \left(\sqrt{\pi_i\pi_j}\right)^{d+2-k}\left(1+\delta_{ij}r^2\right)^{d+1-k} + \delta_{ij}\sum_{k=1}^{d+1}\left(\sqrt{\pi_i\pi_j}\right)^{d+2-k}r^{2k}\left(1+\delta_{ij}r^2\right)^{d+1-k}$$

$$= \sqrt{\pi_i\pi_j}\frac{1-\left(\sqrt{\pi_i\pi_j}\left(1+\delta_{ij}r^2\right)\right)^{d+1}}{1-\sqrt{\pi_i\pi_j}\left(1+\delta_{ij}r^2\right)}$$

$$+ \delta_{ij}\sqrt{\pi_i\pi_j}r^{2(d+1)}\frac{1-\left(\sqrt{\pi_i\pi_j}\left(1+\delta_{ij}r^2\right)r^{-2}\right)^{d+1}}{1-\sqrt{\pi_i\pi_j}\left(1+\delta_{ij}r^2\right)r^{-2}} \tag{21}$$

Since we consider $\sum_{i\in\mathcal{C}_k}\pi_i = 1/K$, the maximum of $\sqrt{\pi_i\pi_j} < 1/4$ for $K = 2$. This implies $\sqrt{\pi_i\pi_j}\left(1+r^2\right) < 1$. Therefore, NTK at $d\to\infty$ is

$$\left(\tilde{\mathbf{\Theta}}_{sym}^{(\infty)}\right)_{ij} = \frac{\sqrt{\pi_i\pi_j}}{1-\sqrt{\pi_i\pi_j}\left(1+\delta_{ij}r^2\right)} \tag{22}$$

Equations (21) and (22) prove the population NTK $\tilde{\mathbf{\Theta}}_{sym}^{(d)}$ and $\tilde{\mathbf{\Theta}}_{sym}^{(\infty)}$ in Theorem 2 and Corollary 1, respectively. $\qquad\square$

### A.2.2 Row Degree Normalized Adjacency $\mathbf{S}_{row}$

The assumption on $\gamma$ in Assumption 2 is only to simplify the expression of population NTK for $\mathbf{S}_{row}$. We derive it without this assumption in the following. We first derive $\mathbf{S}_{row}^k \mathbf{S}_{row}^{kT}$.

$$\mathbf{S}_{row} = \mathbf{D}^{-1}\mathbf{A}$$

$$= \mathbf{D}^{-\frac{1}{2}}\mathbf{D}^{-\frac{1}{2}}\mathbf{A}\mathbf{D}^{-\frac{1}{2}}\mathbf{D}^{+\frac{1}{2}}$$

$$= \mathbf{D}^{-\frac{1}{2}}\mathbf{U}\mathbf{\Lambda}\mathbf{U}^T\mathbf{D}^{+\frac{1}{2}}$$

$$\mathbf{S}_{row}^k = \mathbf{D}^{-\frac{1}{2}}\mathbf{U}\mathbf{\Lambda}^k\mathbf{U}^T\mathbf{D}^{+\frac{1}{2}}$$

$$\mathbf{S}_{row}^k \mathbf{S}_{row}^{kT} = \mathbf{D}^{-\frac{1}{2}}\mathbf{U}\mathbf{\Lambda}^k\mathbf{U}^T\mathbf{D}^{+\frac{1}{2}}\mathbf{D}^{+\frac{1}{2}}\mathbf{U}\mathbf{\Lambda}^k\mathbf{U}^T\mathbf{D}^{-\frac{1}{2}}$$

$$= \mathbf{D}^{-\frac{1}{2}}\mathbf{U}\mathbf{\Lambda}^k\mathbf{U}^T\mathbf{D}\mathbf{U}\mathbf{\Lambda}^k\mathbf{U}^T\mathbf{D}^{-\frac{1}{2}}$$

$$= \left(\mathbf{D}^{-\frac{1}{2}}\mathbf{U}\mathbf{\Lambda}^k\mathbf{U}^T\mathbf{D}^{-\frac{1}{2}}\right)\mathbf{D}^{+\frac{1}{2}}\mathbf{D}\mathbf{D}^{+\frac{1}{2}}\left(\mathbf{D}^{-\frac{1}{2}}\mathbf{U}\mathbf{\Lambda}^k\mathbf{U}^T\mathbf{D}^{-\frac{1}{2}}\right)$$

$$= \left(\hat{\mathbf{U}}\mathbf{\Lambda}^k\hat{\mathbf{U}}^T\right)\mathbf{D}^2\left(\hat{\mathbf{U}}\mathbf{\Lambda}^k\hat{\mathbf{U}}^T\right) \qquad\qquad ; \hat{\mathbf{U}} = \mathbf{D}^{-\frac{1}{2}}\mathbf{U} = \sqrt{\frac{2}{p+q}}\begin{bmatrix}\mathbf{1}_n^T \\ \hat{\mathbf{1}}_n^T\end{bmatrix}_{n\times 2}$$

$$\left(\mathbf{S}_{row}^k \mathbf{S}_{row}^{kT}\right)_{ij} = \begin{cases} \left(1+r^k\right)^2 \lambda + \left(1-r^k\right)^2 \mu & \text{if } i \text{ and } j \in \text{class 1} \\ \left(1+r^k\right)\left(1-r^k\right)(\lambda+\mu) & \text{if } i \text{ and } j \notin \text{same class} \\ \left(1-r^k\right)^2 \lambda + \left(1+r^k\right)^2 \mu & \text{if } i \text{ and } j \in \text{class 2} \end{cases} \;;\; \lambda = \sum_{s=1}^{\frac{n}{2}} \pi_s^2 \;;\; \mu = \sum_{s=\frac{n}{2}+1}^{n} \pi_s^2$$

$$\mathbf{S}_{row}^k \mathbf{S}_{row}^{kT} \overset{\text{matrix not.}}{=} \left[ \begin{array}{c|c} \left(1+r^k\right)^2 \lambda + \left(1-r^k\right)^2 \mu & \left(1+r^k\right)\left(1-r^k\right)(\lambda+\mu) \\ \hline \underbrace{\left(1+r^k\right)\left(1-r^k\right)(\lambda+\mu)}_{\frac{n}{2} \text{ entries}} & \underbrace{\left(1-r^k\right)^2 \lambda + \left(1+r^k\right)^2 \mu}_{\frac{n}{2} \text{ entries}} \end{array} \right]_{n \times n} \tag{23}$$

Note that each block is a constant and independent of individual $\pi_i$. Using (23), NTK in (4) for $i$ and $j$ belonging to class 1 is,

$$\begin{aligned}
\left(\tilde{\boldsymbol{\Theta}}_{row}^{(d)}\right)_{ij} &\overset{(23)}{=} \sum_{k=1}^{d+1} \left(\left(1+r^k\right)^2 \lambda + \left(1-r^k\right)^2 \mu\right) \left(\left(1+r\right)^2 \lambda + \left(1-r\right)^2 \mu\right)^{d+1-k} \\
&= \sum_{k=1}^{d+1} (\lambda+\mu)\left(\left(1+r\right)^2 \lambda + \left(1-r\right)^2 \mu\right)^{d+1-k} + \\
&\quad \sum_{k=1}^{d+1} 2(\lambda-\mu)r^k \left(\left(1+r\right)^2 \lambda + \left(1-r\right)^2 \mu\right)^{d+1-k} + \\
&\quad \sum_{k=1}^{d+1} (\lambda+\mu)r^{2k} \left(\left(1+r\right)^2 \lambda + \left(1-r\right)^2 \mu\right)^{d+1-k} \\
&= (\lambda+\mu)\frac{1 - \left(\left(1+r\right)^2 \lambda + \left(1-r\right)^2 \mu\right)^{d+1}}{1 - \left(\left(1+r\right)^2 \lambda + \left(1-r\right)^2 \mu\right)} + \\
&\quad 2(\lambda-\mu)r^{d+1}\frac{1 - \left(\left(1+r\right)^2 \lambda + \left(1-r\right)^2 \mu\right)^{d+1} r^{-(d+1)}}{1 - \left(\left(1+r\right)^2 \lambda + \left(1-r\right)^2 \mu\right) r^{-1}} + \\
&\quad (\lambda+\mu)r^{2(d+1)}\frac{1 - \left(\left(1+r\right)^2 \lambda + \left(1-r\right)^2 \mu\right)^{d+1} r^{-2(d+1)}}{1 - \left(\left(1+r\right)^2 \lambda + \left(1-r\right)^2 \mu\right) r^{-2}}
\end{aligned} \tag{24}$$

Similarly for $i$ and $j$ in class 2,

$$\begin{aligned}
\left(\tilde{\boldsymbol{\Theta}}_{row}^{(d)}\right)_{ij} &= \sum_{k=1}^{d+1} \left(\left(1-r^k\right)^2 \lambda + \left(1+r^k\right)^2 \mu\right) \left(\left(1-r\right)^2 \lambda + \left(1+r\right)^2 \mu\right)^{d+1-k} \\
&= (\lambda+\mu)\frac{1 - \left(\left(1-r\right)^2 \lambda + \left(1+r\right)^2 \mu\right)^{d+1}}{1 - \left(\left(1-r\right)^2 \lambda + \left(1+r\right)^2 \mu\right)} + \\
&\quad 2(-\lambda+\mu)r^{d+1}\frac{1 - \left(\left(1-r\right)^2 \lambda + \left(1+r\right)^2 \mu\right)^{d+1} r^{-(d+1)}}{1 - \left(\left(1-r\right)^2 \lambda + \left(1+r\right)^2 \mu\right) r^{-1}} + \\
&\quad (\lambda+\mu)r^{2(d+1)}\frac{1 - \left(\left(1-r\right)^2 \lambda + \left(1+r\right)^2 \mu\right)^{d+1} r^{-2(d+1)}}{1 - \left(\left(1-r\right)^2 \lambda + \left(1+r\right)^2 \mu\right) r^{-2}}
\end{aligned} \tag{25}$$

When $i$ and $j$ are in different classes,

$$
\begin{aligned}
\left(\tilde{\boldsymbol{\Theta}}_{row}^{(d)}\right)_{ij} &= \sum_{k=1}^{d+1} \left(1 - r^{2k}\right)(\lambda + \mu)\left(\left(1 - r^2\right)(\lambda + \mu)\right)^{d+1-k} \\
&= \sum_{k=1}^{d+1} (\lambda + \mu)^{d+2-k}\left(1 - r^2\right)^{d+1-k} - r^{2k}(\lambda + \mu)^{d+2-k}\left(1 - r^2\right)^{d+1-k} \\
&= (\lambda + \mu)\frac{1 - (\lambda + \mu)^{d+1}\left(1 - r^2\right)^{d+1}}{1 - (\lambda + \mu)\left(1 - r^2\right)} \\
&\quad - (\lambda + \mu)\, r^{2(d+1)}\frac{1 - (\lambda + \mu)^{d+1}\left(1 - r^2\right)^{d+1} r^{-2(d+1)}}{1 - (\lambda + \mu)\left(1 - r^2\right) r^{-2}}
\end{aligned}
\tag{26}
$$

As $\lambda$ and $\mu < \frac{1}{4}$, $(1 + r)^2 \lambda + (1 - r)^2 \mu < 2\left(1 + r^2\right)\frac{1}{4} < 1$, population NTK $\tilde{\boldsymbol{\Theta}}_{row}$ at $d \to \infty$ is

$$
\left(\tilde{\boldsymbol{\Theta}}_{row}^{(\infty)}\right)_{ij} = 
\begin{cases}
\dfrac{(\lambda + \mu)}{1 - \left((1 + r)^2 \lambda + (1 - r)^2 \mu\right)} & \text{if } i \text{ and } j \in \text{class 1} \\[3mm]
\dfrac{(\lambda + \mu)}{1 - (\lambda + \mu)\left(1 - r^2\right)} & \text{if } i \text{ and } j \in \text{different class} \\[3mm]
\dfrac{(\lambda + \mu)}{1 - \left((1 - r)^2 \lambda + (1 + r)^2 \mu\right)} & \text{if } i \text{ and } j \in \text{class 2}
\end{cases}
\tag{27}
$$

When the assumption on $\gamma$ is introduced, $\exists \gamma \, s.t. \sum_{i=1}^{n} \pi_i^2 \mathbb{1}[\mathcal{C}_i = k] = \gamma \, \forall k$, $\lambda + \mu = 2\gamma$ and $\lambda - \mu = 0$. Hence, equations (24), (25) and (26) of the population NTK $\tilde{\boldsymbol{\Theta}}_{row}^{(d)}$ and (27) of $\tilde{\boldsymbol{\Theta}}_{row}^{(\infty)}$ reduce to the expressions in Theorem 2 and Corollary 1, respectively. $\qquad\square$

### A.2.3    COLUMN NORMALIZED ADJACENCY $\mathbf{S}_{col}$

In this section we derive the population NTK $\tilde{\boldsymbol{\Theta}}_{col}^{(d)}$.

$$
\begin{aligned}
\mathbf{S}_{col} &= \mathbf{A}\mathbf{D}^{-1} \\
&= \mathbf{D}^{+\frac{1}{2}}\mathbf{U}\boldsymbol{\Lambda}\mathbf{U}^T\mathbf{D}^{-\frac{1}{2}} \\
\mathbf{S}_{col}^{k} &= \mathbf{D}^{+\frac{1}{2}}\mathbf{U}\boldsymbol{\Lambda}^{k}\mathbf{U}^T\mathbf{D}^{-\frac{1}{2}} \\
\mathbf{S}_{col}^{k}\mathbf{S}_{col}^{kT} &= \mathbf{D}^{+\frac{1}{2}}\mathbf{U}\boldsymbol{\Lambda}^{k}\mathbf{U}^T\mathbf{D}^{-\frac{1}{2}}\mathbf{D}^{-\frac{1}{2}}\mathbf{U}\boldsymbol{\Lambda}^{k}\mathbf{U}^T\mathbf{D}^{+\frac{1}{2}} \\
&= \left(\tilde{\mathbf{U}}\boldsymbol{\Lambda}^{k}\tilde{\mathbf{U}}^T\right)\mathbf{D}^{-2}\left(\tilde{\mathbf{U}}\boldsymbol{\Lambda}^{k}\tilde{\mathbf{U}}^T\right) \qquad\qquad ;\tilde{\mathbf{U}} = \mathbf{D}^{+\frac{1}{2}}\mathbf{U} = \sqrt{\frac{p + q}{2}}\begin{bmatrix}\boldsymbol{\pi}^T \\ \hat{\boldsymbol{\pi}}^T\end{bmatrix}_{n \times 2} \\
&= n\pi_i \pi_j \left(1 + \delta_{ij} r^{2k}\right)
\end{aligned}
$$

$$
\stackrel{\text{matrix not.}}{\equiv}
\left[
\begin{array}{c|c}
n\pi_i \pi_j \left(1 + r^{2k}\right) & n\pi_i \pi_j \left(1 - r^{2k}\right) \\
\hline
\underbrace{n\pi_i \pi_j \left(1 - r^{2k}\right)}_{\frac{n}{2}\text{ entries}} & \underbrace{n\pi_i \pi_j \left(1 + r^{2k}\right)}_{\frac{n}{2}\text{ entries}}
\end{array}
\right]_{n \times n}
\tag{28}
$$

Therefore, $\tilde{\boldsymbol{\Theta}}_{col}^{(d)}$ is

$$
\begin{aligned}
\left(\tilde{\boldsymbol{\Theta}}_{col}^{(d)}\right)_{ij} &= \sum_{k=1}^{d+1} n\pi_i \pi_j \left(1 + \delta_{ij} r^{2k}\right)\left(n\pi_i \pi_j \left(1 + \delta_{ij} r^2\right)\right)^{d+1-k} \\
&= \sum_{k=1}^{d+1} \left(n\pi_i \pi_j\right)^{d+2-k}\left(1 + \delta_{ij} r^2\right)^{d+1-k} + \delta_{ij}\sum_{k=1}^{d+1}\left(n\pi_i \pi_j\right)^{d+2-k} r^{2k}\left(1 + r^2\right)^{d+1-k} \\
&= n\pi_i \pi_j \left[\frac{1 - \left(n\pi_i \pi_j \left(1 + r^2\right)\right)^{d+1}}{1 - n\pi_i \pi_j \left(1 + r^2\right)} + \delta_{ij} r^{2d+2}\frac{1 - \left(n\pi_i \pi_j \left(1 + r^2\right) r^{-2}\right)^{d+1}}{1 - n\pi_i \pi_j \left(1 + r^2\right) r^{-2}}\right]
\end{aligned}
\tag{29}
$$

Since $\sum_i^n \pi_i = 1$, $\pi_i = \mathcal{O}(\frac{1}{n})$. So, $n\pi_i\pi_j\left(1+r^2\right) < 1$. Therefore, using (29),

$$\left(\tilde{\mathbf{\Theta}}_{col}^{(\infty)}\right)_{ij} = \frac{n\pi_i\pi_j}{1 + n\pi_i\pi_j\left(1 + \delta_{ij}r^2\right)}. \tag{30}$$

Hence, equations (29) and (30) prove the population NTK $\tilde{\mathbf{\Theta}}_{col}^{(d)}$ and $\tilde{\mathbf{\Theta}}_{col}^{(\infty)}$ in Theorem 2 and Corollary 1, respectively. $\qquad\square$

### A.2.4 UNNORMALIZED ADJACENCY $\mathbf{S}_{adj}$

We can rewrite $\mathbf{A}$ as follows,

$$\mathbf{A} = \boldsymbol{\pi}\boldsymbol{\pi}^T \odot \left[\underbrace{\begin{array}{c|c} p & q \\ \hline q & p \end{array}}_{\frac{n}{2}\text{ entries}\quad\frac{n}{2}\text{ entries}}\right]_{n\times n}$$

$$= \begin{bmatrix} \pi_1 & & \\ & \ddots & \\ & & \pi_n \end{bmatrix}_{n\times n} \left[\begin{array}{c|c} p & q \\ \hline q & p \end{array}\right]_{n\times n} \begin{bmatrix} \pi_1 & & \\ & \ddots & \\ & & \pi_n \end{bmatrix}_{n\times n} \tag{31}$$

We consider $\gamma$ assumption for the analysis of unnormalised adjacency to simplify the computation. But the result holds without this assumption.

$$\mathbf{A}^2 \stackrel{(31)}{=} \begin{bmatrix} \pi_1 & & \\ & \ddots & \\ & & \pi_n \end{bmatrix} \left[\begin{array}{c|c} \left(p^2+q^2\right)\gamma & 2pq\gamma \\ \hline 2pq\gamma & \left(p^2+q^2\right)\gamma \end{array}\right] \begin{bmatrix} \pi_1 & & \\ & \ddots & \\ & & \pi_n \end{bmatrix}$$

$$\mathbf{A}^4 = \begin{bmatrix} \pi_1 & & \\ & \ddots & \\ & & \pi_n \end{bmatrix} \left[\begin{array}{c|c} \left(p^4+q^4+6p^2q^2\right)\gamma^3 & \left(4p^3q+4pq^3\right)\gamma^3 \\ \hline \left(4p^3q+4pq^3\right)\gamma^3 & \left(p^4+q^4+6p^2q^2\right)\gamma^3 \end{array}\right] \begin{bmatrix} \pi_1 & & \\ & \ddots & \\ & & \pi_n \end{bmatrix}$$

Note that in the above shown $\mathbf{A}^{2k}$ it is the even powers of binomial expansion of $(p+q)^{2^k}$ for $i, j$ in same class whereas it is the odd powers for $i, j$ not in the same class. We compute the filter $\mathbf{S}_{adj}$ using this fact.

$$\mathbf{S}_{adj} = \frac{1}{n}\mathbf{A}$$

$$\mathbf{S}_{adj}^k = \frac{1}{n^k}\mathbf{A}^k$$

$$\mathbf{S}_{adj}^k\mathbf{S}_{adj}^{kT} = \frac{1}{n^{2k}}\mathbf{A}^{2k}$$

$$= \begin{cases} \pi_i\pi_j\frac{\gamma^{2^k-1}}{n^{2k}}\sum\limits_{l=0}^{2^{k-1}}\binom{2^k}{2l}p^{2^k-2l}q^{2l} & \text{if } i \text{ and } j \in \text{ same class} \\[3mm] \pi_i\pi_j\frac{\gamma^{2^k-1}}{n^{2k}}\sum\limits_{l=0}^{2^{k-1}-1}\binom{2^k}{2l+1}p^{2^k-2l-1}q^{2l+1} & \text{if } i \text{ and } j \in \text{ different class} \end{cases}$$

$$\tilde{\mathbf{\Theta}}_{adj}^{(d)} = \begin{cases} \pi_i\pi_j\sum\limits_{k=1}^{d+1}\frac{\gamma^{2^k+d-k}}{n^{2k}}\left(p^2+q^2\right)^{d+1-k}\sum\limits_{l=0}^{2^{k-1}}\binom{2^k}{2l}p^{2^k-2l}q^{2l} & \text{if } i \text{ and } j \in \text{ same class} \\[3mm] \pi_i\pi_j\sum\limits_{k=1}^{d+1}\frac{\gamma^{2^k+d-k}}{n^{2k}}\left(2pq\right)^{d+1-k}\sum\limits_{l=0}^{2^{k-1}-1}\binom{2^k}{2l+1}p^{2^k-2l-1}q^{2l+1} & \text{if } i \text{ and } j \in \text{ different class} \end{cases}$$

The above form is not simplified as it is not an interesting case where the gap between the two blocks disappears rapidly and $\left(\tilde{\mathbf{\Theta}}_{adj}^{(\infty)}\right)_{ij} = 0$. There is no information in the kernel proving both Theorem 2 and Corollary 1. $\qquad\square$

### A.2.5 Number of Classes $K > 2$

From the above derivation for $K = 2$, it can be seen that once $\mathbf{S}_{sym}^k \mathbf{S}_{sym}^{kT}$ is computed, the population NTK for all the graph convolutions can be derived using it. Therefore, we derive it for $K > 2$ and it suffices to show the conclusions of Theorem 2 and Corollary 1. We denote the vector $\hat{\boldsymbol{\pi}}_{1k}$ with $-\pi_i \forall i \in \left[1, \frac{n}{K}\right]$, $+\pi_i \forall i \in \left[\frac{n(k-1)}{K}, \frac{nk}{K}\right]$ and 0 for the rest. With this definition, $\mathbf{A}$ is

$$\mathbf{A} = \frac{p + (K-1)q}{K} \boldsymbol{\pi}\boldsymbol{\pi}^T + \frac{p-q}{K} \sum_{l=2}^{K} \hat{\boldsymbol{\pi}}_{1l} \hat{\boldsymbol{\pi}}_{1l}^T. \tag{32}$$

$\mathbf{D}$ for $K$ classes is $\frac{p+(K-1)}{K}\mathrm{diag}(\boldsymbol{\pi})$ from (32). We can compute $\mathbf{S}_{sym}$ using $\mathbf{A}$ and $\mathbf{D}$ as follows,

$$\mathbf{S}_{sym} = \mathbf{D}^{-\frac{1}{2}}\mathbf{A}\mathbf{D}^{-\frac{1}{2}}$$

$$= \frac{K}{p+(K-1)q}\mathrm{diag}(\boldsymbol{\pi}^{-\frac{1}{2}})\left(\frac{p+(K-1)q}{K}\boldsymbol{\pi}\boldsymbol{\pi}^T + \frac{p-q}{K}\sum_{l=2}^{K}\hat{\boldsymbol{\pi}}_{1l}\hat{\boldsymbol{\pi}}_{1l}^T\right)\mathrm{diag}(\boldsymbol{\pi}^{-\frac{1}{2}})$$

$$= \boldsymbol{\pi}^{\frac{1}{2}}\boldsymbol{\pi}^{\frac{1}{2}T} + \frac{p-q}{p+(K-1)q}\sum_{l=2}^{K}\hat{\boldsymbol{\pi}}_{1l}^{\frac{1}{2}}\hat{\boldsymbol{\pi}}_{1l}^{\frac{1}{2}T}$$

$$(\mathbf{S}_{sym})_{ij} = \sqrt{\pi_i\pi_j}\left(1 + \delta_{ij}\left(\frac{p-q}{p+(K-1)q}\right)\sum_{l=2}^{K}\frac{K}{l+l^2}\right)$$

$$(\mathbf{S}_{sym}^k)_{ij} = \sqrt{\pi_i\pi_j}\left(1 + \delta_{ij}\left(\frac{p-q}{p+(K-1)q}\right)^k\sum_{l=2}^{K}\frac{K}{l+l^2}\right)$$

$$(\mathbf{S}_{sym}^k\mathbf{S}_{sym}^{kT})_{ij} = \sqrt{\pi_i\pi_j}\left(1 + \delta_{ij}\left(\frac{p-q}{p+(K-1)q}\right)^{2k}\sum_{l=2}^{K}\frac{K}{l+l^2}\right) \tag{33}$$

It is noted that the equation (33) is very much similar to (20) for $K = 2$. The further derivations of the population NTKs $\tilde{\mathbf{\Theta}}$ for all the convolutions are similar and the theoretical results extend without any issues.

### A.3 NTK for GCN with Skip Connections (Corollary 2 and 3)

We observe that the definitions of $\mathbf{G}_i \forall i \in [1, d+1]$ are different for GCN with skip connections from the vanilla GCN. Despite the difference, the definition of gradient with respect to $\mathbf{W}_i$ in (8) does not change as $\mathbf{G}_i$ in the gradient accounts for the change and moreover, there is no new learnable parameter since the input transformation $\mathbf{H}_0 = \mathbf{X}\mathbf{W}_0$ where $(\mathbf{W}_0)_{ij}$ is sampled from $\mathcal{N}(0, 1)$ is not learnable in our setting. Given the fact that the gradient definition holds for GCN with skip connection, the NTK will retain the form from NTK for vanilla GCN as evident from the derivation of NTK for vanilla GCN in Section A.1. The change in $\mathbf{G}_i$ will only affect the co-variance between nodes. Hence, we will derive the co-variance matrix for Skip-PC and Skip-$\alpha$ in the following.

**Skip-PC: Co-variance between nodes.** The co-variance between nodes $u$ and $v$ in $\mathbf{F}_1$ and $\mathbf{F}_i$ are derived below.

$$\mathbb{E}\left[(\mathbf{F}_1)_{uk}(\mathbf{F}_1)_{vk}\right] = \langle(\mathbf{G}_1)_{u.}, (\mathbf{G}_1)_{v.}\rangle$$

$$= \frac{c_\sigma}{h}\langle(\mathbf{S}\sigma_s(\mathbf{H}_0))_{u.}, (\mathbf{S}\sigma_s(\mathbf{H}_0))_{v.}\rangle$$

$$= \frac{c_\sigma}{h}\sum_{k=1}^{h}(\mathbf{S}\sigma_s(\mathbf{H}_0))_{uk}(\mathbf{S}\sigma_s(\mathbf{H}_0))_{vk}$$

$$\overset{h\to\infty}{=} c_\sigma \mathbb{E}\left[(\mathbf{S}\sigma_s(\mathbf{H}_0))_{uk}(\mathbf{S}\sigma_s(\mathbf{H}_0))_{vk}\right] \qquad \text{; law of large numbers}$$

$$= \mathbf{S}_{u.}\tilde{\mathbf{E}}_0\mathbf{S}_{.v}^T \qquad ;\tilde{\mathbf{E}}_0 = c_\sigma \underset{\mathbf{F}\sim\mathcal{N}(\mathbf{0},\mathbf{X}\mathbf{X}^T)}{\mathbb{E}}\left[\sigma_s(\mathbf{F})\sigma_s(\mathbf{F})^T\right]$$

$$= (\mathbf{\Sigma}_1)_{uv} \tag{34}$$

$$\mathbb{E}\left[(\mathbf{F}_i)_{uk}(\mathbf{F}_i)_{vk}\right] = \langle(\mathbf{G}_i)_{u.},(\mathbf{G}_i)_{v.}\rangle$$

$$= \frac{c_\sigma}{h}\langle(\mathbf{S}(\sigma(\mathbf{F}_{i-1})+\sigma_s(\mathbf{H}_0)))_{u.},(\mathbf{S}(\sigma(\mathbf{F}_{i-1})+\sigma_s(\mathbf{H}_0)))_{v.}\rangle$$

$$= \frac{c_\sigma}{h}\sum_{k=1}^{h}(\mathbf{S}\sigma(\mathbf{F}_{i-1})+\mathbf{S}\sigma_s(\mathbf{H}_0))_{uk}(\mathbf{S}\sigma(\mathbf{F}_{i-1})+\mathbf{S}\sigma_s(\mathbf{H}_0))_{vk}$$

$$\overset{h\to\infty}{=} c_\sigma\mathbb{E}\left[(\mathbf{S}\sigma(\mathbf{F}_{i-1})+\mathbf{S}\sigma_s(\mathbf{H}_0))_{uk}(\mathbf{S}\sigma(\mathbf{F}_{i-1})+\mathbf{S}\sigma_s(\mathbf{H}_0))_{vk}\right] \quad \text{; law of large numbers}$$

$$= c_\sigma\Big[\mathbb{E}\left[(\mathbf{S}\sigma(\mathbf{F}_{i-1}))_{uk}(\mathbf{S}\sigma(\mathbf{F}_{i-1}))_{vk}\right]+\mathbb{E}\left[(\mathbf{S}\sigma(\mathbf{F}_{i-1}))_{uk}(\mathbf{S}\sigma_s(\mathbf{H}_0))_{vk}\right]$$

$$+\mathbb{E}\left[(\mathbf{S}\sigma_s(\mathbf{H}_0))_{uk}(\mathbf{S}\sigma(\mathbf{F}_{i-1}))_{vk}\right]+\mathbb{E}\left[(\mathbf{S}\sigma_s(\mathbf{H}_0))_{uk}(\mathbf{S}\sigma_s(\mathbf{H}_0))_{vk}\right]\Big]$$

$$= \mathbf{S}_{u.}\mathbf{E}_{i-1}\mathbf{S}_{.v}^T + c_\sigma\mathbb{E}\left[(\mathbf{S}\sigma(\mathbf{F}_{i-1}))_{uk}(\mathbf{S}\sigma_s(\mathbf{X}\mathbf{W}_0))_{vk}\right]$$

$$+ c_\sigma\mathbb{E}\left[(\mathbf{S}\sigma_s(\mathbf{X}\mathbf{W}_0))_{uk}(\mathbf{S}\sigma(\mathbf{F}_{i-1}))_{vk}\right]$$

$$+ c_\sigma\mathbb{E}\left[\sum_{r=1}^{n}\sum_{s=1}^{n}\mathbf{S}_{ur}\mathbf{S}_{qs}\sigma_s(\mathbf{X}\mathbf{W}_0)_{rk}\sigma_s(\mathbf{X}\mathbf{W}_0)_{sk}\right]$$

$$\overset{(f)}{=} \mathbf{S}_{u.}\mathbf{E}_{i-1}\mathbf{S}_{.v}^T + c_\sigma\mathbf{S}_{u.}\mathbb{E}\left[\sigma_s(\mathbf{X}\mathbf{W}_0)_{rk}\sigma_s(\mathbf{X}\mathbf{W}_0)_{sk}\right]\mathbf{S}_{.v}^T$$

$$= \mathbf{S}_{u.}\mathbf{E}_{i-1}\mathbf{S}_{.v}^T + \mathbf{S}_{u.}\tilde{\mathbf{E}}_0\mathbf{S}_{.v}^T = \mathbf{S}_{u.}\mathbf{E}_{i-1}\mathbf{S}_{.v}^T + (\mathbf{\Sigma}_1)_{uv} = (\mathbf{\Sigma}_i)_{uv} \tag{35}$$

$(f)$: $\mathbb{E}\left[(\mathbf{S}\sigma(\mathbf{F}_{i-1}))_{uk}(\mathbf{S}\sigma_s(\mathbf{X}\mathbf{W}_0))_{vk}\right]$ and $\mathbb{E}\left[(\mathbf{S}\sigma_s(\mathbf{X}\mathbf{W}_0))_{uk}(\mathbf{S}\sigma(\mathbf{F}_{i-1}))_{vk}\right]$ evaluate to 0 by conditioning on $\mathbf{W}_0$ first and rewriting the expectation based on this conditioning. The terms within expectation are independent when conditioned on $\mathbf{W}_0$, and hence it is $\underset{\mathbf{W}_0}{\mathbb{E}}\left[\underset{\mathbf{\Sigma}_{i-1}|\mathbf{W}_0}{\mathbb{E}}[(\mathbf{S}\sigma(\mathbf{F}_{i-1}))_{uk}|\mathbf{W}_0]\underset{\mathbf{\Sigma}_{i-1}|\mathbf{W}_0}{\mathbb{E}}[(\mathbf{S}\sigma_s(\mathbf{X}\mathbf{W}_0))_{vk}|\mathbf{W}_0]\right]$ by taking $h$ in $\mathbf{W}_0$ going to infinity first. Here, $\underset{\mathbf{\Sigma}_{i-1}|\mathbf{W}_0}{\mathbb{E}}[(\mathbf{S}\sigma_s(\mathbf{X}\mathbf{W}_0))_{vk}|\mathbf{W}_0] = 0$.

We get the co-variance matrix for all pairs of nodes $\mathbf{\Sigma}_1 = \mathbf{S}\tilde{\mathbf{E}}_0\mathbf{S}^T$ and $\mathbf{\Sigma}_i = \mathbf{S}\mathbf{E}_{i-1}\mathbf{S}^T + \mathbf{\Sigma}_1$ from (34) and (35).

**Skip-$\alpha$: Co-variance between nodes.** Let $u$ and $v$ be two nodes and the co-variance between $u$ and $v$ in $\mathbf{F}_1$ and $\mathbf{F}_i$ are derived below.

$$\mathbb{E}\left[(\mathbf{F}_1)_{uk}(\mathbf{F}_1)_{vk}\right] = \langle(\mathbf{G}_1)_{u.},(\mathbf{G}_1)_{v.}\rangle$$

$$= \frac{c_\sigma}{h}\sum_{k=1}^{h}((1-\alpha)\mathbf{S}\sigma_s(\mathbf{H}_0)+\alpha\sigma_s(\mathbf{H}_0))_{uk}((1-\alpha)\mathbf{S}\sigma_s(\mathbf{H}_0)+\alpha\sigma_s(\mathbf{H}_0))_{vk}$$

$$\overset{h\to\infty}{=} c_\sigma\mathbb{E}\left[((1-\alpha)\mathbf{S}\sigma_s(\mathbf{H}_0)+\alpha\sigma_s(\mathbf{H}_0))_{uk}((1-\alpha)\mathbf{S}\sigma_s(\mathbf{H}_0)+\alpha\sigma_s(\mathbf{H}_0))_{vk}\right]$$

$$= c_\sigma\Big[(1-\alpha)^2\mathbb{E}\left[(\mathbf{S}\sigma_s(\mathbf{H}_0))_{uk}(\mathbf{S}\sigma_s(\mathbf{H}_0))_{vk}\right]$$

$$+ (1-\alpha)\alpha\left(\mathbb{E}\left[(\mathbf{S}\sigma_s(\mathbf{H}_0))_{uk}(\sigma_s(\mathbf{H}_0))_{vk}\right]+\mathbb{E}\left[(\mathbf{S}\sigma_s(\mathbf{H}_0))_{vk}(\sigma_s(\mathbf{H}_0))_{uk}\right]\right)$$

$$+ \alpha^2\mathbb{E}\left[(\sigma_s(\mathbf{H}_0))_{uk}(\sigma_s(\mathbf{H}_0))_{vk}\right]$$

$$= (1-\alpha)^2\mathbf{S}_{u.}\tilde{\mathbf{E}}_0\mathbf{S}_{.v}^T+(1-\alpha)\alpha\left(\mathbf{S}_{u.}\left(\tilde{\mathbf{E}}_0\right)_{.v}+\left(\tilde{\mathbf{E}}_0\right)_{u.}\mathbf{S}_{.v}^T\right)+\alpha^2\left(\tilde{\mathbf{E}}_0\right)_{uv} = (\mathbf{\Sigma}_1)_{uv} \tag{36}$$

$$\mathbb{E}\left[(\mathbf{F}_i)_{uk}(\mathbf{F}_i)_{vk}\right] = \langle (\mathbf{G}_i)_{u.}, (\mathbf{G}_i)_{v.} \rangle$$

$$= \frac{c_\sigma}{h} \sum_{k=1}^{h} \left((1-\alpha)\mathbf{S}\sigma(\mathbf{F}_{i-1}) + \alpha\sigma_s(\mathbf{H}_0)\right)_{uk} \left((1-\alpha)\mathbf{S}\sigma(\mathbf{F}_{i-1}) + \alpha\sigma_s(\mathbf{H}_0)\right)_{vk}$$

$$\overset{h\to\infty}{=} c_\sigma \mathbb{E}\left[\left((1-\alpha)\mathbf{S}\sigma(\mathbf{F}_{i-1}) + \alpha\sigma_s(\mathbf{H}_0)\right)_{uk} \left((1-\alpha)\mathbf{S}\sigma(\mathbf{F}_{i-1}) + \alpha\sigma_s(\mathbf{H}_0)\right)_{vk}\right]$$

$$= c_\sigma \Big[ (1-\alpha)^2 \mathbb{E}\left[(\mathbf{S}\sigma(\mathbf{F}_{i-1}))_{uk}(\mathbf{S}\sigma(\mathbf{F}_{i-1}))_{vk}\right] + \alpha^2 \mathbb{E}\left[(\sigma_s(\mathbf{H}_0))_{uk}(\sigma_s(\mathbf{H}_0))_{vk}\right]$$

$$+ (1-\alpha)\alpha\left(\mathbb{E}\left[(\mathbf{S}\sigma(\mathbf{F}_{i-1}))_{uk}(\sigma_s(\mathbf{H}_0))_{vk}\right] + \mathbb{E}\left[(\sigma_s(\mathbf{H}_0))_{uk}(\mathbf{S}\sigma(\mathbf{F}_{i-1}))_{vk}\right]\right)\Big]$$

$$\overset{(g)}{=} (1-\alpha)^2 \mathbf{S}_{u.}\mathbf{E}_{i-1}\mathbf{S}_{.v}^T + \alpha^2 \left(\tilde{\mathbf{E}}_0\right)_{uv} = (\mathbf{\Sigma}_i)_{uv} \tag{37}$$

$(g)$: same argument as $(f)$ in derivation of $\mathbf{\Sigma}_i$ in Skip-PC.

We get the co-variance matrix for all pairs of nodes $\mathbf{\Sigma}_1 = (1-\alpha)^2\mathbf{S}\tilde{\mathbf{E}}_0\mathbf{S}^T + \alpha(1-\alpha)\left(\mathbf{S}\tilde{\mathbf{E}}_0 + \tilde{\mathbf{E}}_0\mathbf{S}^T\right) + \alpha^2\tilde{\mathbf{E}}_0$ and $\mathbf{\Sigma}_i = (1-\alpha)^2\mathbf{S}\mathbf{E}_{i-1}\mathbf{S}^T + \alpha^2\tilde{\mathbf{E}}_0$ from (36) and (37).

### A.4 Theorem 3: Population NTK $\tilde{\mathbf{\Theta}}$ for Skip-PC

NTK at depth $d$, $\mathbf{\Theta}_{PC}^{(d)}$ for Skip-PC with linear activations is

$$\mathbf{\Theta}_{PC}^{(d)} = \sum_{k=1}^{d+1} \left(\mathbf{S}^k\mathbf{S}^{kT} + \mathbf{S}\mathbf{S}^T\right) \odot \left(\mathbf{S}\mathbf{S}^T\right)^{\odot d+1-k}$$

$$= \sum_{k=1}^{d+1} \underbrace{\mathbf{S}^k\mathbf{S}^{kT} \odot \left(\mathbf{S}\mathbf{S}^T\right)^{\odot d+1-k}}_{I} + \underbrace{\left(\mathbf{S}\mathbf{S}^T\right)^{\odot d+2-k}}_{II} \tag{38}$$

In (38), $I$ is NTK without skip connection and $II$ is computed for $\mathbf{S}_{row}$ and $\mathbf{S}_{sym}$ as follows.

Computing $II$ for population NTK $\tilde{\mathbf{\Theta}}^{(d)}$ for $\mathbf{S}_{sym}$: for nodes $i$ and $j$,

$$\sum_{k=1}^{d+1} \left(\mathbf{S}_{sym}\mathbf{S}_{sym}^T\right)_{ij}^{\odot d+2-k} = \sum_{k=1}^{d+1} \left(\sqrt{\pi_i\pi_j}\left(1+\delta_{ij}r^2\right)\right)^{d+2-k}$$

$$= \sqrt{\pi_i\pi_j}\left(1+\delta_{ij}r^2\right) \frac{1 - \left(\sqrt{\pi_i\pi_j}\left(1+\delta_{ij}r^2\right)\right)^{d+1}}{1 - \sqrt{\pi_i\pi_j}\left(1+\delta_{ij}r^2\right)}$$

$$\overset{d\to\infty}{=} \frac{\sqrt{\pi_i\pi_j}\left(1+\delta_{ij}r^2\right)}{1 - \sqrt{\pi_i\pi_j}\left(1+\delta_{ij}r^2\right)} \tag{39}$$

It converges to (39) as $d \to \infty$ since $\sqrt{\pi_i\pi_j}\left(1+\delta_{ij}r^2\right) < 1$ according to our setup. Therefore, using (39) and (22) we get the population NTK $\tilde{\mathbf{\Theta}}_{PC,sym}^{(\infty)}$ for Skip-PC at $d \to \infty$,

$$\left(\tilde{\mathbf{\Theta}}_{PC,sym}^{(\infty)}\right)_{ij} = \frac{\sqrt{\pi_i\pi_j}\left(2+\delta_{ij}r^2\right)}{1 - \sqrt{\pi_i\pi_j}\left(1+\delta_{ij}r^2\right)},$$

hence deriving Theorem 3. $\qquad\square$

Similarly, computing $II$ for $\mathbf{S}_{row}$ without assumption on $\gamma$, $i$ and $j$ in class 1,

$$\sum_{k=1}^{d+1} \left(\mathbf{S}_{row}\mathbf{S}_{row}^T\right)_{ij}^{\odot d+2-k} = \sum_{k=1}^{d+1} \left((1+r)^2\lambda + (1-r)^2\mu\right)^{d+2-k}$$

$$= \left((1+r)^2\lambda + (1-r)^2\mu\right) \frac{1 - \left((1+r)^2\lambda + (1-r)^2\mu\right)^{d+1}}{1 - \left((1+r)^2\lambda + (1-r)^2\mu\right)}$$

$$\overset{d\to\infty}{=} \frac{(1+r)^2\,\lambda + (1-r)^2\,\mu}{1 - \left((1+r)^2\,\lambda + (1-r)^2\,\mu\right)} \tag{40}$$

For $i$ and $j$ in class 2,

$$\sum_{k=1}^{d+1} \left(\mathbf{S}_{row}\mathbf{S}_{row}^T\right)_{ij}^{\odot d+2-k} = \left((1-r)^2\,\lambda + (1+r)^2\,\mu\right) \frac{1 - \left((1-r)^2\,\lambda + (1+r)^2\,\mu\right)^{d+1}}{1 - \left((1-r)^2\,\lambda + (1+r)^2\,\mu\right)}$$

$$\overset{d\to\infty}{=} \frac{(1-r)^2\,\lambda + (1+r)^2\,\mu}{1 - \left((1-r)^2\,\lambda + (1+r)^2\,\mu\right)} \tag{41}$$

For $i$ and $j$ in different class,

$$\sum_{k=1}^{d+1} \left(\mathbf{S}_{row}\mathbf{S}_{row}^T\right)_{ij}^{\odot d+2-k} = \left(1-r^2\right)\left(\lambda+\mu\right) \frac{1 - \left(\left(1-r^2\right)\left(\lambda+\mu\right)\right)^{d+1}}{1 - \left(1-r^2\right)\left(\lambda+\mu\right)}$$

$$\overset{d\to\infty}{=} \frac{\left(1-r^2\right)\left(\lambda+\mu\right)}{1 - \left(1-r^2\right)\left(\lambda+\mu\right)} \tag{42}$$

Therefore, the population NTK $\tilde{\Theta}_{\alpha,row}^{(\infty)}$ with $\gamma$ assumption is obtained by substituting $\lambda + \mu = 2\gamma$ and $\lambda - \mu = 0$ in (40), (41) and (42) .

$$\tilde{\Theta}_{\alpha,row}^{(\infty)} = \frac{2\gamma(2 + \delta_{ij}r^2)}{1 - 2\gamma\left(1 + \delta_{ij}r^2\right)},$$

hence deriving Theorem 3 $\qquad\qquad\qquad\qquad\qquad\qquad\qquad\qquad\qquad\qquad\qquad\qquad\square$.

### A.5 THEOREM 4: POPULATION NTK $\tilde{\Theta}$ FOR SKIP-$\alpha$

We expand $\mathbf{\Sigma}_1$ and $\mathbf{\Sigma}_k$ of Skip-$\alpha$ first to derive the population NTK.

$$\mathbf{\Sigma_1} = (1-\alpha)^2\,\mathbf{SS}^T + \alpha\,(1-\alpha)\left(\mathbf{S} + \mathbf{S}^T\right) + \alpha^2\mathbf{I}_n$$

$$\mathbf{\Sigma_k} = (1-\alpha)^2\,\mathbf{S\Sigma_{k-1}S^T} + \alpha^2\mathbf{I_n}$$

$$= (1-\alpha)^{2k}\,\mathbf{S}^k\mathbf{S}^{kT} + \alpha\,(1-\alpha)^{2k-1}\,\mathbf{S}^{k-1}\left(\mathbf{S}+\mathbf{S}^T\right)\mathbf{S}^{k-1^T} + \alpha^2\sum_{l=0}^{k-1}(1-\alpha)^{2l}\,\mathbf{S}^l\mathbf{S}^{lT} \tag{43}$$

Exact NTK of depth $d$ for Skip-$\alpha$ is expanded using the above as follows.

$$\mathbf{\Theta}_\alpha^{(d)} = \sum_{k=1}^{d+1}\mathbf{\Sigma_k} \odot \left(\mathbf{SS^T}\right)^{\odot \mathbf{d+1-k}}$$

$$= \underbrace{\sum_{k=1}^{d+1}(1-\alpha)^{2k}\,\mathbf{S}^k\mathbf{S}^{kT} \odot \left(\mathbf{SS}^T\right)^{\odot d+1-k}}_{I} +$$

$$\underbrace{\sum_{k=1}^{d+1}\alpha\,(1-\alpha)^{2k-1}\,\mathbf{S}^{k-1}\left(\mathbf{S}+\mathbf{S}^T\right)\mathbf{S}^{k-1^T} \odot \left(\mathbf{SS}^T\right)^{\odot d+1-k}}_{II} +$$

$$\underbrace{\sum_{k=1}^{d+1}\alpha^2\sum_{l=0}^{k-1}(1-\alpha)^{2l}\,\mathbf{S}^l\mathbf{S}^{lT} \odot \left(\mathbf{SS}^T\right)^{\odot d+1-k}}_{III} \tag{44}$$

We compute $I$, $II$ and $III$ of (44) for population NTK $\tilde{\Theta}_\alpha^{(\infty)}$ using $\mathbf{S}_{sym}$ focusing on $d \to \infty$.

$$I_{ij} = (1-\alpha)^{2(d+1)} \sqrt{\pi_i \pi_j} \left[ \frac{1 - \left( \sqrt{\pi_i \pi_j} \left(1 + \delta_{ij} r^2\right) (1-\alpha)^{-2} \right)^{d+1}}{1 - \sqrt{\pi_i \pi_j} \left(1 + \delta_{ij} r^2\right) (1-\alpha)^{-2}} + \right.$$
$$\left. r^{2(d+1)} \frac{1 - \left( \sqrt{\pi_i \pi_j} \left(1 + \delta_{ij} r^2\right) r^{-2} (1-\alpha)^{-2} \right)^{d+1}}{1 - \sqrt{\pi_i \pi_j} \left(1 + \delta_{ij} r^2\right) r^{-2} (1-\alpha)^{-2}} \right] \overset{d \to \infty}{=} 0 \qquad (45)$$

$$II = \alpha \sum_{k=1}^{d+1} (1-\alpha)^{2k-1} \, 2 \mathbf{S}_{sym}^{2k-1} \odot \left( \mathbf{S}_{sym} \mathbf{S}_{sym}^T \right)^{\odot d+1-k} \qquad ; \mathbf{S}_{sym} = \mathbf{S}_{sym}^T$$

$$II_{ij} = 2\alpha \sum_{k=1}^{d+1} \left[ (1-\alpha)^{2k-1} \left( \sqrt{\pi_i \pi_j} \right)^{d+k} \left(1 + \delta_{ij} r^2\right)^{d+1-k} + \right.$$
$$\left. (1-\alpha)^{2k-1} \left( \sqrt{\pi_i \pi_j} \right)^{d+k} \left(1 + \delta_{ij} r^2\right)^{d+1-k} r^{2k-1} \delta_{ij} \right]$$

$$= 2\alpha \left[ \left( \sqrt{\pi_i \pi_j} (1-\alpha) \right)^{2d+1} \frac{1 + \left( \left( \sqrt{\pi_i \pi_j} \right)^{-1} (1-\alpha)^{-2} \left(1 + \delta_{ij} r^2\right) \right)^{d+1}}{1 + \left( \left( \sqrt{\pi_i \pi_j} \right)^{-1} (1-\alpha)^{-2} \left(1 + \delta_{ij} r^2\right) \right)} + \right.$$
$$\left. \delta_{ij} \left( r \sqrt{\pi_i \pi_j} (1-\alpha) \right)^{2d+1} \frac{1 + \left( \left( \sqrt{\pi_i \pi_j} \right)^{-1} (r(1-\alpha))^{-2} \left(1 + \delta_{ij} r^2\right) \right)^{d+1}}{1 + \left( \left( \sqrt{\pi_i \pi_j} \right)^{-1} (r(1-\alpha))^{-2} \left(1 + \delta_{ij} r^2\right) \right)} \right] \overset{d \to \infty}{=} 0 \quad (46)$$

$$III = \alpha^2 \sum_{k=1}^{d+1} \left[ \sum_{l=0}^{k-1} (1-\alpha)^{2l} \, \mathbf{S}_{sym}^l \mathbf{S}_{sym}^{lT} \right] \odot \left( \mathbf{S}_{sym} \mathbf{S}_{sym}^T \right)^{\odot d+1-k}$$

$$III_{ij} = \alpha^2 \sum_{k=1}^{d+1} \left[ \sum_{l=0}^{k-1} (1-\alpha)^{2l} \left(1 + \delta_{ij} r^{2l}\right) \sqrt{\pi_i \pi_j} \right] \left( \left(1 + \delta_{ij} r^2\right) \sqrt{\pi_i \pi_j} \right)^{d+1-k}$$

$$= \alpha^2 \sqrt{\pi_i \pi_j} \sum_{k=1}^{d+1} \left( \frac{1}{1 - (1-\alpha)^2} + \frac{\delta_{ij}}{1 - (1-\alpha)^2 r^2} \right) \left( \left(1 + \delta_{ij} r^2\right) \sqrt{\pi_i \pi_j} \right)^{d+1-k} +$$
$$\left( -\frac{(1-\alpha)^{2k}}{1 - (1-\alpha)^2} - \delta_{ij} \frac{(1-\alpha)^{2k} r^{2k}}{1 - (1-\alpha)^2 r^2} \right) \left( \left(1 + \delta_{ij} r^2\right) \sqrt{\pi_i \pi_j} \right)^{d+1-k}$$

$$= \alpha^2 \sqrt{\pi_i \pi_j} \left( \left( \frac{1}{1 - (1-\alpha)^2} + \frac{\delta_{ij}}{1 - (1-\alpha)^2 r^2} \right) \frac{1 - \left( \sqrt{\pi_i \pi_j} \left(1 + \delta_{ij} r^2\right) \right)^{d+1}}{1 - \sqrt{\pi_i \pi_j} \left(1 + \delta_{ij} r^2\right)} \right.$$
$$- \frac{(1-\alpha)^{2(d+1)}}{1 - (1-\alpha)^2} \frac{1 - \left( \sqrt{\pi_i \pi_j} \left(1 + \delta_{ij} r^2\right) (1-\alpha)^{-2} \right)^{d+1}}{1 - \left( \sqrt{\pi_i \pi_j} \left(1 + \delta_{ij} r^2\right) (1-\alpha)^{-2} \right)}$$
$$\left. - \frac{\delta_{ij} (1-\alpha)^{2(d+1)} r^{2(d+1)}}{1 - (1-\alpha)^2 r^2} \frac{1 - \left( \sqrt{\pi_i \pi_j} \left(1 + \delta_{ij} r^2\right) (1-\alpha)^{-2} r^{-2} \right)^{d+1}}{1 - \sqrt{\pi_i \pi_j} \left(1 + \delta_{ij} r^2\right) (1-\alpha)^{-2} r^{-2}} \right)$$
$$\overset{d \to \infty}{=} \frac{\alpha^2 \sqrt{\pi_i \pi_j}}{1 - \sqrt{\pi_i \pi_j} \left(1 + \delta_{ij} r^2\right)} \left( \frac{1}{1 - (1-\alpha)^2} + \frac{\delta_{ij}}{1 - (1-\alpha)^2 r^2} \right) \qquad (47)$$

Therefore the population NTK as $d \to \infty$ is obtained by combining (45), (46) and (47).

$$\left( \tilde{\Theta}_{\alpha,sym}^{(\infty)} \right)_{ij} = \frac{\alpha^2 \sqrt{\pi_i \pi_j}}{1 - \sqrt{\pi_i \pi_j} \left(1 + \delta_{ij} r^2\right)} \left( \frac{1}{1 - (1-\alpha)^2} + \frac{\delta_{ij}}{1 - (1-\alpha)^2 r^2} \right),$$

proving Theorem 4. □

We now compute $I$, $II$ and $III$ for population NTK $\tilde{\Theta}_\alpha^{(\infty)}$ using $\mathbf{S}_{row}$. For nodes $i$ and $j$ in class 1:

$$I_{ij} = (\lambda + \mu)(1-\alpha)^{2(d+1)} \frac{1 - \left((1+r)^2\lambda + (1-r)^2\mu\right)^{d+1}(1-\alpha)^{-2(d+1)}}{1 - \left((1+r)^2\lambda + (1-r)^2\mu\right)(1-\alpha)^{-2}} +$$

$$2(\lambda-\mu)(1-\alpha)^{2(d+1)}r^{d+1}\frac{1 - \left((1+r)^2\lambda + (1-r)^2\mu\right)^{d+1}r^{-(d+1)}(1-\alpha)^{-2(d+1)}}{1 - \left((1+r)^2\lambda + (1-r)^2\mu\right)r^{-1}(1-\alpha)^{-2}} +$$

$$(\lambda+\mu)(1-\alpha)^{2(d+1)}r^{2(d+1)}\frac{1 - \left((1+r)^2\lambda + (1-r)^2\mu\right)^{d+1}r^{-2(d+1)}(1-\alpha)^{-2(d+1)}}{1 - \left((1+r)^2\lambda + (1-r)^2\mu\right)r^{-2}(1-\alpha)^{-2}}$$

$$\overset{d\to\infty}{=} 0 \tag{48}$$

Similarly for nodes $i$ and $j$ in class 2 and different classes $I_{ij} = 0$ as $d \to \infty$. Likewise, $II_{ij} = 0$ as $d \to \infty$ for any $i$ and $j$. This is similar to $\mathbf{S}_{sym}$.

For $i$ and $j$ in class 1,

$$III = \alpha^2\sum_{k=1}^{d+1}\left[\sum_{l=0}^{k-1}(1-\alpha)^{2l}\mathbf{S}_{row}^l\mathbf{S}_{row}^{lT}\right] \odot \left(\mathbf{S}_{row}\mathbf{S}_{row}^T\right)^{\odot d+1-k}$$

$$III_{ij} = \alpha^2\sum_{k=1}^{d+1}\left[\sum_{l=0}^{k-1}(1-\alpha)^{2l}\left((\lambda+\mu)(1+r^{2l}) + (\lambda-\mu)2r^l\right)\right]\left((1+r)^2\lambda + (1-r)^2\mu\right)^{d+1-k}$$

$$= \alpha^2\sum_{k=1}^{d+1}\left[(\lambda+\mu)\left(\frac{1-(1-\alpha)^{2k}}{1-(1-\alpha)^2} + \frac{1-(1-\alpha)^{2k}r^{2k}}{1-(1-\alpha)^2r^2}\right) + (\lambda-\mu)\frac{1-(1-\alpha)^{2k}r^k}{1-(1-\alpha)^2r}\right] \times$$

$$\left((1+r)^2\lambda + (1-r)^2\mu\right)^{d+1-k}$$

$$= \alpha^2\left(\frac{\lambda+\mu}{1-(1-\alpha)^2} + \frac{\lambda+\mu}{1-(1-\alpha)^2r^2} + \frac{\lambda-\mu}{1-(1-\alpha)^2r}\right)\sum_{k=1}^{d+1}\left((1+r)^2\lambda + (1-r)^2\mu\right)^{d+1-k} -$$

$$\alpha^2\left(\frac{\lambda+\mu}{1-(1-\alpha)^2}\right)\sum_{k=1}^{d+1}(1-\alpha)^{2k}\left((1+r)^2\lambda + (1-r)^2\mu\right)^{d+1-k} -$$

$$\alpha^2\left(\frac{\lambda+\mu}{1-(1-\alpha)^2r^2}\right)\sum_{k=1}^{d+1}(1-\alpha)^{2k}r^{2k}\left((1+r)^2\lambda + (1-r)^2\mu\right)^{d+1-k} -$$

$$\alpha^2\left(\frac{\lambda-\mu}{1-(1-\alpha)^2r}\right)\sum_{k=1}^{d+1}(1-\alpha)^{2k}r^k\left((1+r)^2\lambda + (1-r)^2\mu\right)^{d+1-k}$$

$$= \alpha^2\left(\frac{\lambda+\mu}{1-(1-\alpha)^2} + \frac{\lambda+\mu}{1-(1-\alpha)^2r^2} + \frac{\lambda-\mu}{1-(1-\alpha)^2r}\right)\frac{1-\left((1+r)^2\lambda + (1-r)^2\mu\right)^{d+1}}{1-\left((1+r)^2\lambda + (1-r)^2\mu\right)} -$$

$$\alpha^2\left(\frac{\lambda+\mu}{1-(1-\alpha)^2}\right)(1-\alpha)^{2(d+1)}\frac{1-\left((1+r)^2\lambda + (1-r)^2\mu\right)^{d+1}(1-\alpha)^{-2(d+1)}}{1-\left((1+r)^2\lambda + (1-r)^2\mu\right)(1-\alpha)^{-2}} -$$

$$\alpha^2\left(\frac{\lambda+\mu}{1-(1-\alpha)^2r^2}\right)((1-\alpha)r)^{2(d+1)}\frac{1-\left((1+r)^2\lambda + (1-r)^2\mu\right)^{d+1}((1-\alpha)r)^{-2(d+1)}}{1-\left((1+r)^2\lambda + (1-r)^2\mu\right)((1-\alpha)r)^{-2}} -$$

$$
\alpha^2 \left( \frac{\lambda - \mu}{1 - (1-\alpha)^2 r} \right) \left( (1-\alpha)^2 r \right)^{(d+1)} \frac{1 - \left( (1+r)^2 \lambda + (1-r)^2 \mu \right)^{d+1} \left( (1-\alpha)^2 r \right)^{-(d+1)}}{1 - \left( (1+r)^2 \lambda + (1-r)^2 \mu \right) \left( (1-\alpha)^2 r \right)^{-1}}
$$

$$
\stackrel{d \to \infty}{=} \alpha^2 \left( \frac{\lambda + \mu}{1 - (1-\alpha)^2} + \frac{\lambda + \mu}{1 - (1-\alpha)^2 r^2} + \frac{\lambda - \mu}{1 - (1-\alpha)^2 r} \right) \frac{1}{1 - \left( (1+r)^2 \lambda + (1-r)^2 \mu \right)}
$$

$$(49)$$

Similarly for $i$ and $j$ in class 2,

$$
III_{ij} \stackrel{d \to \infty}{=} \alpha^2 \left( \frac{\lambda + \mu}{1 - (1-\alpha)^2} + \frac{\lambda + \mu}{1 - (1-\alpha)^2 r^2} - \frac{\lambda - \mu}{1 - (1-\alpha)^2 r} \right) \frac{1}{1 - \left( (1-r)^2 \lambda + (1+r)^2 \mu \right)}
$$

$$(50)$$

For $i$ and $j$ in different classes,

$$
III_{ij} \stackrel{d \to \infty}{=} \alpha^2 \left( \frac{\lambda + \mu}{1 - (1-\alpha)^2} - \frac{\lambda + \mu}{1 - (1-\alpha)^2 r^2} \right) \frac{1}{1 - (1-r^2)(\lambda + \mu)}
$$

$$(51)$$

Thus, applying $\gamma$ assumption to (49), (50) and (51) the population NTK $\tilde{\Theta}_{\alpha,row}^{(\infty)}$ as $d \to \infty$ is,

$$
\left( \tilde{\Theta}_{\alpha,row}^{(\infty)} \right)_{ij} = \frac{2\gamma\alpha^2}{1 - 2\gamma\left(1 + \delta_{ij}r^2\right)} \left( \frac{1}{1 - (1-\alpha)^2} + \frac{\delta_{ij}}{1 - (1-\alpha)^2 r^2} \right),
$$

hence proving Theorem 4. □

## B  EMPIRICAL ANALYSIS

### B.1  EXPERIMENTAL DETAILS OF FIGURE 1

We use the code for GCN without skip connections from github1(Kipf & Welling, 2017) and skip connection from github2(Chen et al., 2020). The following hyperparameters are used for GCN without skip connections: learning rate is $0.01$, weight decay is $5e-4$, hidden layer width is $64$ and epochs is $500, 1500, 2000$ for depths $2, 4, 8$ respectively. For the skip connections, we used GCNII model, same parameters as vanilla GCN with $\alpha = 0.1$. The performance is averaged over 5 runs.

### B.2  COMPARISON OF GCN AND NTK

Although it is theoretically clear that the infinite width assumption should not affect the observations made on performance of GCN with $\mathbf{S}_{sym}$ and $\mathbf{S}_{row}$ in Figure 1, we illustrate the same using graph NTK. Figure 5 shows that the observation is seen in graph NTK as well, thus supporting our theoretical argument.

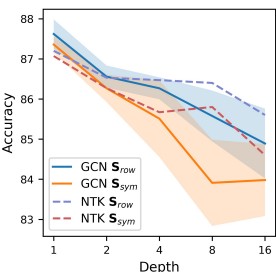

Figure 5: Comparison of the accuracy of a trained finite width GCN and the corresponding NTK.

### B.3 NUMERICAL VALIDATION FOR DC-SBM FOR VANILLA GCN AND SKIP-$\alpha$

**Experimental Details**. For the experiments, we fix the size of the sampled graphs to $n = 1000$, $p = 0.8$ and $q = 0.1$ for homophily DC-SBM, $p = 0.1$ and $q = 0.8$ for heterophily DC-SBM and $p = q = 1$ for core-periphery DC-SBM. $\pi$ is sampled uniformly $[0, 1]$ for homophily and heterophily, and $\pi_i \sim \text{Unif}(0.5, 1) \forall i \in core$ and $\pi_i \sim \text{Unif}(0, 0.5) \forall i \in periphery$ for core-periphery DC-SBM.

**Illustration of impact of depth in Vanilla GCN using Homophily DC-SBM**. We show the impact of depth in Vanilla GCN using homophily DC-SBM Figure 6. The DC-SBM is shown in the first column and columns 2 and 3 show the exact NTK for depth=1 and 8 for symmetric and row normalization, respectively. The plots clearly illustrate the complete loss of class information in symmetric normalization with depth (column 2). While the prevalence of block difference has decresed in row normalization over depth (column 3), the block/community structure is still retained. Thus showing the strong representation power of $\mathbf{S}_{row}$.

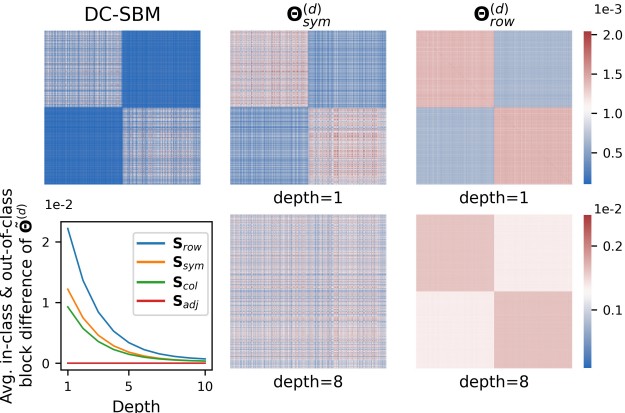

Figure 6: Numerical validation of Theorem 2 using DC-SBM shown in the first plot of column 1. Columns 2 and 3 illustrate the exact NTKs of depth=1 and 8 for $\mathbf{S}_{sym}$ and $\mathbf{S}_{row}$, respectively. Second plot in column 1 shows the average gap between in-class and out-of-class blocks from theory.

**Illustration of $\mathbf{S}_{col}$ and $\mathbf{S}_{adj}$ in Vanilla GCN using Homophily DC-SBM**. We extend the experiments on numerical validation for random graphs using vanilla GCN described in Section 3.2 to column normalized adjacency $\mathbf{S}_{col}$ and unnormalized adjacency $\mathbf{S}_{adj}$ here. We use the same setup described in Section 3.2 and Figure 7 illustrates the results. We observe that even for depth 1 both the convolutions are influenced by the degree correction and there is no class information in the kernels for higher depth. Thus, this validates the theoretical result in Theorem 2.

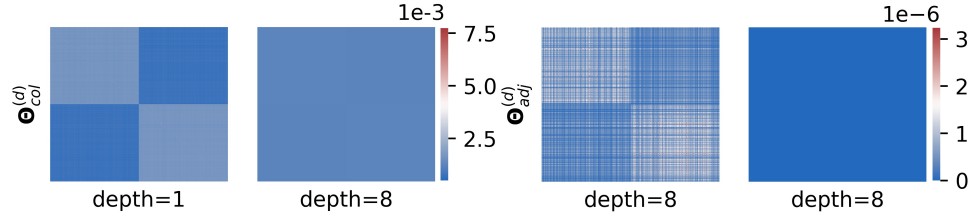

Figure 7: Numerical validation of DC-SBM for Vanilla GCN. The first two heatmaps show the exact NTK $\Theta^{(d)}$ for column normalized adjacency convolution $\mathbf{S}_{col}$ and the other two for unnormalized adjacency $\mathbf{S}_{adj}$ for depths $d = 1$ and 8.

**Illustration of impact of depth in Skip-PC and Skip-$\alpha$ using Homophily DC-SBM**. We present a complementary result to Section 4.3 here. We use the same setting as described in Section 4.3 and

plot the exact NTKs of depths 1 and 8 for symmetric and row normalization. Figure 8 shows the results for Skip-PC and we observe that the gap between in-class and out-of-class blocks decreases for both $\mathbf{S}_{row}$ and $\mathbf{S}_{sym}$ with depth, but the class information is still retained for larger depth and the gap doesn't vanish. Between $\mathbf{S}_{row}$ and $\mathbf{S}_{sym}$, the heatmaps show that $\mathbf{S}_{row}$ retains the block structure better than $\mathbf{S}_{sym}$ and is devoid of the influence of the degree corrections.

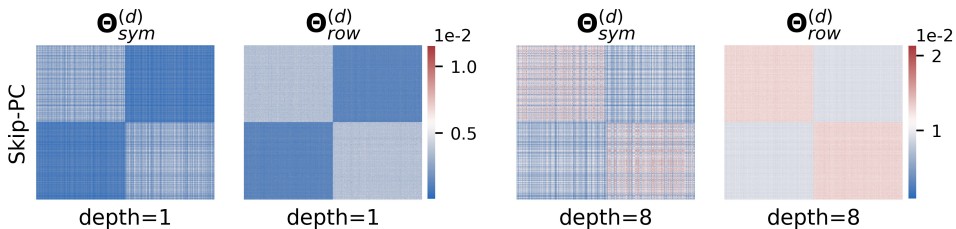

Figure 8: Numerical validation of DC-SBM for Skip-PC showing the exact NTKs $\mathbf{\Theta}^{(d)}$ for $\mathbf{S}_{sym}$ and $\mathbf{S}_{row}$ for depths 1 and 8.

In the case of Skip-$\alpha$, we use $\alpha = 0.1$ to obtain the result illustrated in Figure 9. Similar conclusions are derived from the experiment. Although we consider $\mathbf{X}\mathbf{X}^T = \mathbf{I}_n$ for Skip-$\alpha$ which fundamentally relies on the feature information to interpolate, the results are still meaningful and demonstrate the theoretical findings.

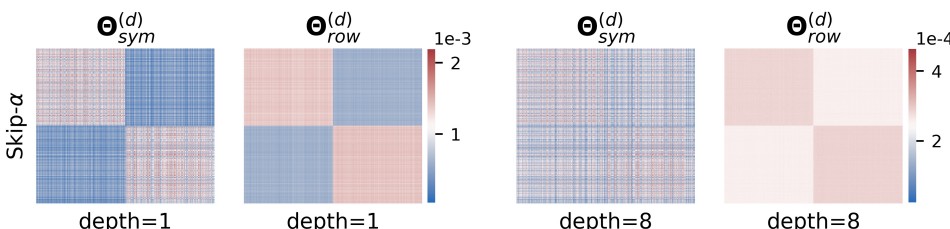

Figure 9: Numerical validation of DC-SBM for Skip-$\alpha$ showing the exact NTKs $\mathbf{\Theta}^{(d)}$ for $\mathbf{S}_{sym}$ and $\mathbf{S}_{row}$ for depths 1 and 8.

**Numerical Validation of Core-Periphery DC-SBM**. In this section, we validate the two scenarios discussed in Section 5 - core-periphery without community structure and core-periphery with community structure. For the firsr case, we consider core-periphery DC-SBM with $n/4$ nodes as core and the rest as periphery as shown in the first heatmap of Figure 10. We plot the exact NTKs of depth 2 for symmetric and row normalization using Vanilla GCN as shown in the second and third heatmaps of Figure 10. This clearly demonstrates the theoretical result presented in Corollary 4 where the symmetric normalization exhibits the graph structure and the row normalization is a constant kernel.

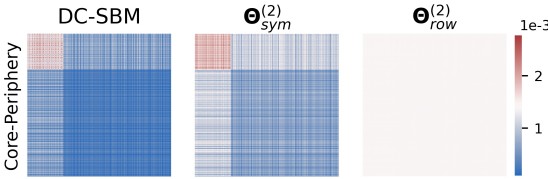

Figure 10: Numerical validation of Core-Periphery DC-SBM showing the exact NTKs $\mathbf{\Theta}^{(d)}$ for $\mathbf{S}_{sym}$ and $\mathbf{S}_{row}$ for depth 2.

In the second setting, we consider two communities of equal size $n/2$ with core-periphery in each, and the link probabilities between cores of the communities is higher than core-periphery or periphery-periphery of the two communities as shown in the first heatmap of Figure 11. The exact NTKs of symmetric and row normalization are illustrated in the second and third heatmaps of Figure 11 where we see that row normalization retains the community structure again.

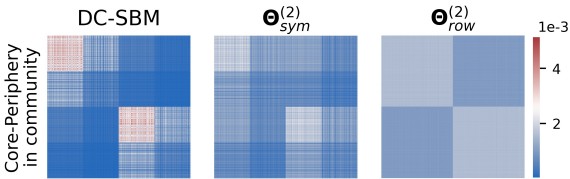

Figure 11: Numerical validation of Core-Periphery DC-SBM with community structure showing the exact NTKs $\Theta^{(d)}$ for $\mathbf{S}_{sym}$ and $\mathbf{S}_{row}$ for depth 2.

### B.4 EXPERIMENTS ON REAL DATASET: CORA

**Orthonormal Feature $\mathbf{XX}^T = \mathbf{I}_n$ Assumption**. In this section, we present additional experiments on Cora. Since our theory assumed orthonormal features $\mathbf{XX}^T = \mathbf{I}_n$, we validate it experimentally in similar setup described in Section 6. Figure 12 shows the result for $\mathbf{S}_{sym}$ and $\mathbf{S}_{row}$ for depth 1 and 8. The conclusions derived from real setting hold here as well and shows $\mathbf{S}_{row}$ preserves the class information better than $\mathbf{S}_{sym}$.



Figure 12: Evaluation on Cora with $\mathbf{XX}^T = \mathbf{I}_n$ for $\mathbf{S}_{sym}$ and $\mathbf{S}_{row}$ for depths 1 and 8.

**ReLU GCN.** We present the result for ReLU GCN in this section. Figure 13 shows the result where the conclusions derived in Section 6 holds very well. Additionally, we plot the average in-class and out-of-class block difference in the case of vanilla GCN (line plots in first row of Figure 13), we observe that the average in-class and out-of-class block difference degrades with depth for each class in Cora, showing the negative impact of depth which aligns well with the theoretical result.

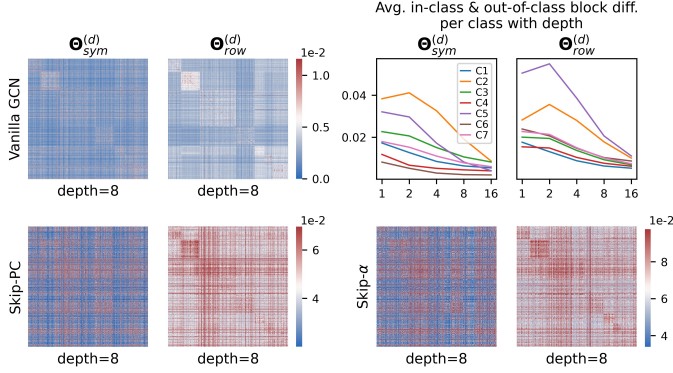

Figure 13: Evaluation on Cora dataset. Heatmaps show results of vanilla GCN and the decrease in class separability with depth for $\mathbf{S}_{sym}$ and $\mathbf{S}_{row}$. Last two show NTKs of Skip-PC where a min and max threshold of 30 and 70 percentile is set for better visualization. Skip-$\alpha$ results in Appendix.

Another experimental study is to understand how easy it is to learn the classes that showed good in-class and out-of-class gap preservation from the above experiment. The line plot in Figure 13 shows class $C2$ and $C5$ are well represented by both $\mathbf{S}_{sym}$ and $\mathbf{S}_{row}$. To study how well this holds in the trained GCN, we considered depth 4 vanilla GCN with ReLU activations and used the same hyperparameters mentioned in Section B.1. The results are shown in Figure 14 where we observe that $C2$ and $C5$ are well learnt. On the other hand, other classes that showed small gap are also well learnt by the trained GCN. This needs further investigation as it has to do with the data split and some classes are poorly represented in the training data, for instance $C6$. Thus, we leave it for further analysis.

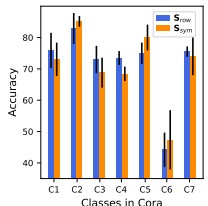

Figure 14: Class wise performance of trained GCN of depth 4.

**Linear GCN.** We present the result for linear GCN with the same setup as described in Section 6 to check the goodness of our theory. The results are illustrated in Figure 15 where we observe that the theory holds very well for linear GCN than ReLU GCN. The class information is better preserved in $\mathbf{S}_{row}$ than $\mathbf{S}_{sym}$ especially for higher depth in the case of both GCN with and without skip connections. All the conclusions derived in the main section hold here as well.

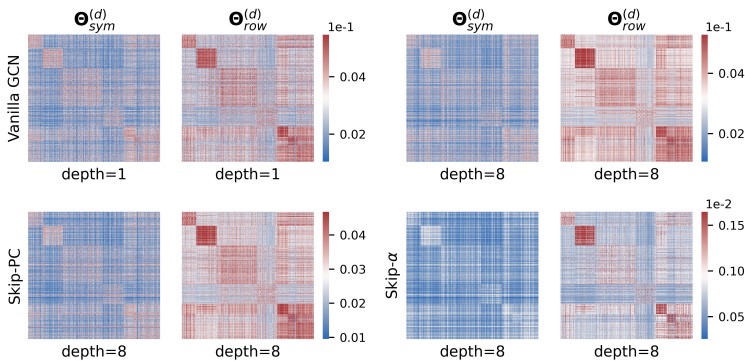

Figure 15: Evaluation on Cora using linear GCN. First row shows the results for vanilla GCN for depths 1 and 8. Second row shows the result for Skip-PC and Skip-$\alpha$ for depth 8.

### B.5 EXPERIMENTS ON REAL DATASET: CITESEER

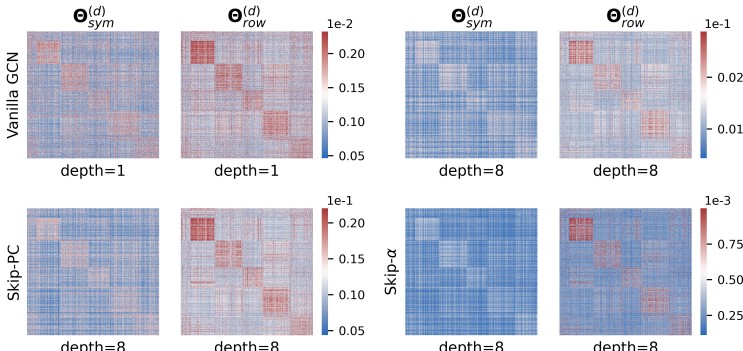

Figure 16: Evaluation on Citeseer dataset using linear GCN. First row shows the results for vanilla GCN for depths 1 and 8. Second row shows the result for Skip-PC and Skip-$\alpha$ for depth 8.

In this section, we validate our theoretical findings on Citeseer without much of the assumptions. We consider multi-class node classification ($K = 6$) using GCN with linear activations and relax the orthonormal feature condition, so $\mathbf{X}\mathbf{X}^T \neq \mathbf{I}_n$. The NTKs for vanilla GCN, GCN with Skip-PC and Skip-$\alpha$ for depths $d = 1, 2, 4, 8, 16$ are computed and Figure 16 illustrates the results. All the observations made in Section 6 hold here as well and clear blocks emerge for $\mathbf{S}_{row}$ making it the preferable choice as suggested in the theory.

