# OpenReview forum: "Representation Power of Graph Convolutions : Neural Tangent Kernel Analysis"
_ICLR.cc/2023/Conference — Submitted to ICLR 2023_

### Official Review · Reviewer_vkDP · 2022-10-17

**Confidence:** 4
**Correctness:** 4
**Technical Novelty And Significance:** 2
**Empirical Novelty And Significance:** Not applicable
**Recommendation:** 3

**Clarity, Quality, Novelty And Reproducibility:**

Novelty:
The author(s) claims that the paper is the first theoretical proof of this topic. However, after the proof, they fail to provide new knowledge about the GCNs field.

Quality:
The proof looks legit and correct.

Clarity:
A few pronouns are not well defined, thus bringing confusion. For example, in the second sentence in the abstract, it is not clear what the authors refer to by "its".

Reproducibility:
Datasets and implementation are provided in the supplementary. The empirical part of this paper is not critical though.

**Strength And Weaknesses:**

Strength:
This paper provides a theoretical analysis of why row normalization is better than symmetric normalization, which is absent from current research. Thus, it provides insight and potential research direction.

Weakness:
1. Graph neural tangent kernel is presented by others before.
2. The observations are discovered previously by others empirically.
3. The underlying reasons for the observations have been discussed or at least inferred somewhere else.
4. No further proposal is made based on the proof.

**Summary Of The Paper:**

This paper utilizes the graph neural tangent kernel to theoretically prove the superiority of row normalization in a semi-supervised node classification setting. It is proven that row normalization can preserve the class structure better, and work better at reducing over-smoothing, especially when combined with skip connections,

**Summary Of The Review:**

By theoretically proving that row normalization and skip connection work better in semi-supervised node classification tasks, this paper serves as a good suggestion for how to implement GCNs. However, it lacks a further discussion on what we can do differently from the current existing approaches.

---

> ### Author Response · Authors · 2022-11-17
> **Response to reviewer's comments**
>
> The reviewer focuses on the lack of a new method. However, we believe that not every paper in ICLR needs to be method oriented. A good amount of theoretical works are very much essential to precisely model and unravel the perceived ‘black box’ nature of deep learning models. This is an important step to further ML algorithmic development as well.
>
> We would respectively argue that the points under weakness mentioned by the reviewer cannot be considered as weaknesses for the following reasons:
>
> ### 1. Graph neural tangent kernel is presented by others before.
> To the best of our knowledge, GNTK as presented in [Du et al.] is the only result with regards to NTK in the graph setting, however, for a different learning problem (supervised graph classification) unlike our semi-supervised node classification setting. If this is not the case we ask the reviewer to point us to the relevant work. As discussed in the paper, the GNTK for this setting and for skip connection are not readily existent and had to be derived. Moreover, the objective of our work is not to simply present the GNTK, but to use it as a tool to analyze the representation power of graph convolutions, the role of depth and skip connections in GCNs which is novel. Thus, we disagree with this point being stated as a weakness.
>
> ### 2. The observations are discovered previously by others empirically.
> We would like to stress upon the fact that this precisely is the **motivation** for our theoretical analysis. Given that it is observed empirically in some works mentioned and also showcased experimentally (Figure 1) in the paper, we focused on theoretically explaining this observation using GNTK as a tool. Moreover, with experiments it is only possible to observe certain characteristics in a specific setting (particular data set, model, optimization, etc.). But, with theoretical analysis, we can obtain the exact characterization of the learning problem under mild assumptions. It is quite common in deep learning (theory) literature that a phenomenon is empirically observed in several works before a theoretical explanation is available.  Hence, it is certainly not a weakness of the paper.
>
> ### 3. The underlying reasons for the observations have been discussed or at least inferred somewhere else.
> To the best of our knowledge, the effect of the graph convolutions particularly in GCNs are not discussed or theoretically derived in any prior works, especially the under performance of the popular choice of the convolution (symmetric normalization) over the row normalization. If this is not the case, we would ask the reviewer to give references to the relevant works.
>
> ### 4. No further proposal is made based on the proof.
> We highlight that the main objective of the paper is to provide theoretical answers to the empirical observation, that is, theoretically understand the representation power of different graph convolutions, thereby extending the analysis to understand the role of depth and skip connections in GCNs. Thus, we disagree with the reviewer on this point, since our analysis establishes strong reasons to use row normalization as the convolution in the presence of class structure, and shallow depth in GCNs with a detailed understanding of the role of skip connections from the perspective of GNTK. We have also discussed this in detail in the Conclusion section of the paper.
>
> The reviewer had mentioned that this paper provides no new knowledge about GCNs in novelty and no further discussion on what can be done differently in summary. We disagree with the reviewer on both the points, since we have precisely stated and validated our contribution in theoretically understanding several aspects of the GCNs in the paper.
>
> ### References
> [Du et al.] "Graph neural tangent kernel: Fusing graph neural networks with graph kernels." Advances in neural information processing systems 32 (2019).

---

> > ### Comment · Reviewer_vkDP · 2022-11-25
> > **Response to the authors**
> >
> > Thanks for your response and I appreciate your effort in justifying your own work. However, I still couldn't see the meaning of this work since it didn't offer any new information. I firmly believe that you can develop it better in the future. Good luck.

---

### Official Review · Reviewer_CyLz · 2022-10-20

**Confidence:** 1
**Clarity, Quality, Novelty And Reproducibility:** I am not in good position to review
**Correctness:** 4
**Technical Novelty And Significance:** 4
**Empirical Novelty And Significance:** 4
**Recommendation:** 5

**Strength And Weaknesses:**

I am not in good position to review

**Summary Of The Paper:**

I am not in good position to review

**Summary Of The Review:**

I am not in good position to review

---

### Official Review · Reviewer_aSCc · 2022-10-24

**Confidence:** 2
**Correctness:** 2
**Technical Novelty And Significance:** 2
**Empirical Novelty And Significance:** 2
**Recommendation:** 3

**Clarity, Quality, Novelty And Reproducibility:**

Clarity

This paper is well-written. I do not have any difficulty understanding the main points of the paper.


Quality

I have a question about the problem setting of the theoretical analysis because it looks somewhat artificial. This paper claimed the usefulness of row-normalized adjacency matrices by comparing the behavior of GNTKs (Theorem 2). This result strongly depends on the assumption that the underlying SBM is degree-correlated. I expected the authors to show why it is reasonable to assume the graph-generating process used in this paper.

I could not see why Theorem 2 shows the usefulness of row-normalized adjacency matrices. This paper claimed the superiority of row-normalized adjacency matrices on the ground that "only $\tilde{\Theta}_{\mathrm{row}}$ exhibits a block structure unaffected by the degree correction $\pi$, and the gap is determined by $r^2$ and $d$". However, I could not see why the independence from $\pi$ implies the superiority of row-normalized adjacency matrices.

The relationship between theoretical analysis and empirical evaluation was not very clear to me.
According to the paper, we have an empirical fact that GNNs with row-normalized adjacency matrices perform well. Therefore, I expect the theoretical analysis's assumptions are empirically reasonable. However, this paper did not discuss the justification of the assumptions. Also, numerical evaluations considered the situation where the assumption of theoretical analysis does not hold (even approximately). Instead, this paper claimed the practical superiority of row-normalized adjacency matrices without assumptions. Therefore, it seems to me that it implies that the assumptions are superfluous and do not reflect real-world situations.
Therefore, I could not see what the numerical evaluations wanted to show and how they were related to theoretical analysis.


Novelty

As far as I know, no paper has given a theoretical explanation for the usefulness of row-normalized adjacency matrices in GNNs. So in that sense, this paper is novel.
On the other hand, I question whether the problem of this paper is sufficiently significant because, to the best of my knowledge, the superiority of row-normalized adjacency matrices is not well-known. This paper referred Wang et al., 2018, Wang & Leskovec, 2020, and Ragesh et al., 2021 as existing studies of GNNs using row-normalized NNs. However, it is difficult to say that they support the significance of the research question of this paper:
- Wang et al., 2018 certainly claimed the usefulness of row-normalized adjacency matrices. However, they discussed it in the context of adversarial attacks. The relevance of the node prediction considered in this issue is not apparent.
- In Wang & Leskovec, 2020, row-normalized adjacency is introduced because of the analogy with label propagation. However, they did not compare it with a symmetrized adjacency matrix was not made.
- Ragesh et al., 2021 used three types of NNs: raw, row-normalized, and symmetrized matrices. However, it did not state which one was the best.
Therefore, I have question about the significance of the research question this paper addressed.


Reproducibility

The code and datasets for the numerical experiments are provided. It allows readers to see the details of the implementation. In addition, instructions for reproduction are provided. Although I have not run the code, I expect that it is possible to reproduce the same experiments as the authors.

**Details Of Ethics Concerns:**

N.A.

**Strength And Weaknesses:**

Strengths
- Few studies have investigated the impact of the design choice of adjacency matrices as a graph convolution operator.

Weakness
- I have a question about the significance of the research question that this paper addressed (see Novelty section).
- The theoretical analysis needs to discuss the validity of the assumptions more (see Quality section).
- The interpretation of the results of the theoretical analysis is questionable
- It is desirable to clarify the relationship between the theoretical analysis and the numerical experiments more.

**Summary Of The Paper:**

This paper attempted to explain the superiority of row-normalized adjacency matrices as the graph convolution operator for node classification in GNNs from the viewpoint of graph neural tangent kernels (GNTK). This paper derived GNTKs for GNNs with finite and infinite layers (and with an infinite number of units) for various adjacency matrices when the underlying graph is a populated Degree-corrected SBM (DC-SBM), and the input matrix is orthogonal. By comparing the GNTKs, this paper showed that the row-normalized adjacency matrix is more capable of identifying nodes than the other design choices (symmetrized, raw, and column-normalized adjacency matrices). In addition, this paper considered GNTK for GNNs with initial residuals and showed that GNTK could identify nodes even with an infinite number of layers, provably avoiding over-smoothing.

**Summary Of The Review:**

Although the choice of the adjacency matrix is one of the design choices, the significance of the research question is questionable, considering the line of previous studies. Also, unless I missed it, the reason for making that assumption about the theoretical analysis is not discussed qualitatively or experimentally, and its validity is questionable.

---

> ### Comment · Reviewer_aSCc · 2022-11-17
> **Post Discussion Period Comment**
>
> So far, the authors do not have any comment. I want to keep my score.

---

> > ### Comment · Reviewer_aSCc · 2022-11-17
> > **I have mistakenly recognized the reviewing process.**
> >
> > I mistakenly understood the reviewing process and thought that I should determine the final score now.I understand it is not now and that the current period is the discussion phase between authors and reviewers.
> >
> > I am sorry for the confusion. I am looking forward to the authors' responses.

---

> > > ### Author Response · Authors · 2022-11-17
> > > **Looking forward to a fruitful discussion**
> > >
> > > Thank you for your interest.
> > > We mistook the discussion period till November 18th as the time to update the paper and the discussions with the reviewers to happen even in stage 2 since the ‘Reviewing process’ in the ICLR webpage states it slightly differently, hence the delay. Hope to have a fruitful discussion. Thanks.

---

> ### Author Response · Authors · 2022-11-17
> **Response to reviewer's concerns (DC-SBM assumption, theorem 2) [1/2]**
>
> The reviewer questions the significance of the paper as the superiority of row normalization is not shown in many prior works. We note that a large number of applications use symmetric normalization because it was suggested in popular GCN works [Kipf & Welling] and there exists no rigorous theoretical study on different convolutions in GCNs before our analysis. Furthermore, while the official Tensorflow implementation of GCN [Kipf & Welling] uses symmetric normalization, the official pytorch implementation of GCN [Kipf & Welling] uses row normalization as default (See https://github.com/tkipf/pygcn/blob/master/pygcn/utils.py#L39 ). This implies every work that adopts this implementation of GCN assumes symmetric normalization is being applied to the adjacency, but actually uses row normalization in practice.
> This shows the lack of clarity on the graph convolutions just from the experiments, and hence, the best choice of parametrization for deep neural networks needs to be theoretically supported rather than hit and trial based experimental guidelines using specific handpicked datasets.
>
> We clarify the concerns raised by the reviewer in the following.
>
> ### 1. Degree Corrected Stochastic Block Model assumption
> For any theoretical analysis, one needs to make minimal assumptions on the underlying data distribution and/or model. In our case, we considered DC-SBM (not as restrictive as SBM) for the data distribution as this is one of the standard settings for the theoretical analysis of graph learning problems [Abbe]. Moreover, some standard practical methods are motivated based on SBM [Klicpera et al].
>
> * **Reason for considering degree correction**: in the case of simple SBM, row normalization and symmetric normalization show no difference in expectation regardless of the learning problem (this can also be inferred from Theorem 2 by ignoring degree correction, that is, $\mathbf{\pi}=c$, $c$ is a constant). Hence, we considered degree correction to differentiate different convolutions in the population version, and our analysis brought out how each convolution influences the representation power of the graph convolutions.
> * **Generalizing the analysis**: Our work sets up the framework for understanding different convolutions in general. Thus, it can be easily extended to any data generating process as well as any new convolutions developed in the future. We have addressed this in the current analysis and also added new experimental results (Figures 2, 10, 11) in the updated version of the paper (see also our response 1 for reviewer *mpKm*).
>
> ### 2. Clarification on Theorem 2 - why independence from $\pi$ implies superiority of row-normalized adjacency matrices
> The class structure of the DC-SBM model is determined by $p$ and $q$ only. $\pi$ adds degree correction to the nodes which determine the connections between nodes but not the class labeling. From the GNTK, we observe that all convolution operators other than row normalization have the dependence on $\pi$, which means that the class structure is influenced by $\pi$ as well. Whereas, row normalization has no influence of $\pi$ implying that the class structure is retained as the underlying DC-SBM (exhibiting block structure similar to the original DC-SBM), making it better than others. We clarified this in the updated version of the paper.
>
> ### References
> [Kipf & Welling] Semi-supervised classification with graph convolutional networks. In International Conference on Learning Representations (ICLR), 2017.
>
> [Abbe] Community Detection and Stochastic Block Models. The Journal of Machine Learning Research 18.1 (2017): 6446-6531.
>
> [Klicpera et al] Diffusion Improves Graph Learning. NeurIPS (2019)

---

> > ### Author Response · Authors · 2022-11-17
> > **Response to reviewer's concerns (theory vs experiments, significance of our work) [2/2]**
> >
> > ### 3. Theoretical analysis and empirical evaluation
> > As we already clarified in 1, the theoretical analysis is not possible in the real world setting, and hence we made the following assumptions,
> > 1. **Infinite width analysis**: this consideration enables us to analyze the behavior without the effect of network parameters or optimization as it has a concise analytic expression.
> > 2. **Linear GCN**: the derived NTK in Theorem 1 is generic for any activation function. However, analyzing with ReLU activation has two problems: (a) it is difficult to analyze deeper GCNs since the expression will get hard to simplify, (b) even for shallow setting, it would be difficult to arrive at an interpretable expression such as we derived for linear GCN (degree correction destroys the class structure in all convolutions except row normalization), since the effect will be observed along with the non-linearity leading to more complexity.
> > 3. **DC-SBM**: as explained in 1, we need to make data distributional assumptions to analyze any behavior theoretically and DC-SBM is a very standard model.
> >
> > In addition, we address the concerns regarding the deviation in activation function in experiments on real data (ReLU) and the theory (linear) by now plotting the results for linear networks in Figure 4. We moved the corresponding plot for ReLU to the appendix (Figure 13). The main statements about class structure preservation stay the same however. This again underlines the fact that our analysis, while done under simplifying assumptions extends to real world data and networks as we. Hence, our theoretical setting is not very restrictive and empirically we show that it captures real world behavior.
> >
> > ### 4. Significance of the theoretical analysis of different graph convolutions
> > As discussed before, the choice of the graph convolution in GCNs is not backed theoretically. Symmetric normalization is popularly used as it was suggested in the GCN paper [Kipf & Welling]. However, as mentioned before their pytorch implementation considers row normalization and since pytorch is one of the top platforms for deep learning, there is a considerable number of papers that assumed symmetric normalization but applied row normalization in practice. The fact that there is no consensus even between implementations on the preferred method further illustrates that a purely empirical approach is not sufficient to determine the optimal convolution operator. Hence, it is important to theoretically support the choice to improve the applicability of GCNs and understand when and why it learns a better graph representation.
> >
> > In conclusion, with experiments (such as the direct comparison of filters shown in Figure 1), it is only possible to observe certain characteristics in a specific setting (particular data, model, parameters, etc.). However, with a theoretical analysis, we can obtain the exact characterization of the learning problem under mild assumptions (in our case, we derived that given the existence of community structure in the data, GCNs with row normalization outperform GCNs with symmetric normalization by preserving the class structures).
> >
> > ### References
> > [Kipf & Welling] Semi-supervised classification with graph convolutional networks. In International Conference on Learning Representations (ICLR), 2017.

---

> > > ### Comment · Reviewer_aSCc · 2022-11-18
> > > **Thank you for your response**
> > >
> > > I thank the authors for replying to my review comment. I will carefully read the response and update my comment accordingly.

---

> > > ### Comment · Reviewer_aSCc · 2022-12-02
> > > **Response**
> > >
> > > I thank the authors for carefully reading my comments and responding to them.
> > >
> > > **Implementation of PyTorch GCN**
> > >
> > > I agree that it makes sense to give a theoretical explanation for the choice between the two since it gives us a guide for the design choice.
> > > Also, I understand that the official implementation of the PyTorch version of GCN uses a row-normalized adjacency matrix. However, with that being said, it does not necessarily imply the superiority of the row-normalized adjacency matrix over the symmetric one.
> > >
> > > **1. Degree Corrected Stochastic Block Model assumption**
> > >
> > > I agree that making assumptions about the data generation process for theoretical analysis is a must because theoretical analysis is impossible without them. The question is what assumptions are made and how they are justified.
> > >
> > > > in the case of simple SBM, row normalization and symmetric normalization show no difference [...]. Hence, we considered degree correction to differentiate different convolutions in the population version, [...].
> > >
> > > This point is what I was concerned about. There could be a natural question of whether the degree-corrected problem set-up is deliberately chosen to conclude that the row-normalized matrix is theoretically preferable. That is why it is preferable to have the justification of the assumptions of theoretical analysis, including degree-corrected such as experimental verifications.
> > >
> > > **2. Clarification on Theorem 2**
> > >
> > > I thank the authors for the additional explanations. However, I am afraid I do not think the independence of $\Theta_{\mathrm{row}}$ from $\pi$ implies the superiority of the row-normalized matrix. To me, a naive way to evaluate the ease of class separability would be to compare the difference between in-class and out-class components of the NTK matrix $\Theta$, as shown in Figure 2, regardless of the dependence on $\pi$. In this sense, the result shown in Figure 2 Column 4 can be evidence of the row-normalized matrix's superiority (I have a question about the configuration used in this figure. See Question below). It is true that the separability of $\Theta_{\mathrm{row}}$ does not depend on $\pi$. However, which of $\Theta_{\mathrm{row}}$ and other NTKs can depend on the value of $\pi$.
> > >
> > > **3. Theoretical analysis and empirical evaluation**
> > >
> > > See Q1 for the validity of the assumptions for the theoretical analysis.
> > > Regarding the comparison of row-normalized and symmetric matrices, I would say it is difficult to judge which is better from the heatmaps in Figures 4 and 13 because we can see the block structure in the symmetric matrix results. On the other hand, the line plot in Figure 13 shows quantitatively that the row-normalized matrix has a larger gap between the in-class and out-class than the symmetric matrix, affirmative to the superiority of the row-normalized matrix in this setting.
> > >
> > > **4. Significance of the theoretical analysis of different graph convolutions**
> > >
> > > See the "Implementation of PyTorch GCN" section for the implementation of GCN and Q1 for the assumptions of the theoretical analysis.
> > >
> > > **Question**
> > >
> > > In Figure 2 Column 4, are the values of $p$, $q$, and $\pi$ chosen as described in the Experimental Details in Section B.3?

---

> > > > ### Author Response · Authors · 2022-12-05
> > > > **Clarification**
> > > >
> > > > We thank the reviewer for the response.
> > > > ### Superiority of row normalization
> > > > Our justification based on PyTorch implementation is only to strengthen the case. We primarily rely on the empirical evidence in Figure 1 and [Wang et al] to say that row normalization is better than symmetric normalization. This may also appear handpicked since it is done only on the popular Cora dataset. To alleviate that concern, we provide the empirical results for Citeseer and Cornell datasets below evaluated using GNTK of the linear GCN,
> > > >
> > > > | Depth | Citeseer row | Citeseer sym | Cornell row | Cornell sym|
> > > > | -------- |   -------- |   -------- |   -------- |   -------- |
> > > > | 1 |  **70.25** |   69.05 |   **57.83** |   50.60 |
> > > > | 2 |  **69.20** |   65.50 |   **52.51** |   51.81 |
> > > > | 4 |  **68.35** |   65.30 |   49.40 |   **50.40** |
> > > > | 8 |  **68.30** |   65.90 |   47.00 |   **50.60** |
> > > >
> > > > This shows that row normalization is typically better than symmetric normalization and achieves the best performance.
> > > >
> > > > ### DC-SBM assumption
> > > > The theoretical setup is influenced by the empirical observation and not the other way around. Since empirically there is a difference between row and symmetric normalization, we chose a model that enables this study.
> > > >
> > > > ### $p,q, \pi$ values
> > > > Yes, the values are chosen as described in B.3

---

> > > > > ### Comment · Reviewer_aSCc · 2022-12-12
> > > > > **Response**
> > > > >
> > > > > I thank the authors for responsing my comments. I think I deepen my understanding about the paper.
> > > > >
> > > > > **Superiority of row normalization**
> > > > >
> > > > > I thank the authors for sharing the additional numerical results about comparing the row and symmetric matrices. I understand that the row-normalized matrix is better in Cora and Citeseer, especially when the depth is small.
> > > > >
> > > > > **DC-SBM assumption**
> > > > >
> > > > > I understand that the authors claimed that the theory is justified because it can explain the empirical evidence well. However, I do not think that drawing a conclusion that aligns with the empirical evidence is not sufficient to justify theoretical analyses, especially their assumptions. There is a possibility that we impose unnecessarily strong assumptions. Also, if assumptions do not reflect the real-world setting, the theory can deviate from what is going on under the hood practically. That is why I suggested the justification of the assumptions.

---

### Official Review · Reviewer_nvrd · 2022-10-25

**Confidence:** 4
**Correctness:** 2
**Technical Novelty And Significance:** 2
**Empirical Novelty And Significance:** 2
**Recommendation:** 3

**Clarity, Quality, Novelty And Reproducibility:**

The paper is mostly clear and well-written, but certain passages are difficult to understand. A non-exhaustive list:

- Assumption 2 and the corresponding remark are unclear; in particular, why does this assumption "enable(s) the computation of analytic expression for the population NTK instead of the expected NTK?"
- Theorem 2 is hard to parse, and the discussion following it is not obvious from the theorem. Either simplify the equations, or conduct a more detailed discussion.

**Details Of Ethics Concerns:**

N/A.

**Strength And Weaknesses:**

Strengths:

- The problem addressed by the paper---understanding the effect of design choices such row/symmetric normalization, depth, and residual connections---is timely and important.
- The empirical results in Figure 3 (left) are convincing in showing the effect of skip connections on class separability in node classification problems.

Weaknesses:

- The motivation of the paper is somewhat ill-defined. The motivation for analyzing the effect of depth comes from the observation, in Figure 1, that the performance of GCNs with symmetrically normalized adjacency matrices degrades faster with depth than that of GCNs with row normalized adjacency matrices. However, if the best accuracies in this figure are achieved for 2 layers, and the difference between symmetric and row normalization in the 2-layer case is small, why are we concerned with what happens for a larger (or even infinite) number of layers?
- The modeling assumptions restrict the applicability of the analysis, as it is likely that the SBM model used to conduct the NTK analysis and the specific task (community detection/separability of communities) are biased towards the row normalized adjacency matrix (also known as the random walk matrix). It is well known that community detection methods based on random walks (e.g., node embeddings such as DeepWalk; spectral methods based on the non-backtracking operator) fare better than methods based on symmetric aggregations. Moreover, not every node classification problem can be modeled as detecting communities, especially when considering heterophilous graphs. This is for instance suggested by [r1] below, where it is noted that the row-normalized adjacency is more limited in the amount of topological information that it carries.
- Although the result in Theorem 2 agrees with the numerical results, i.e., it explains why the row normalized adjacency does better than the symmetrically normalized adjacency in practice, the result itself gives no intuition as to why that is the case, and an interpretation is not given by the authors.
- Although the NTK analysis is only accurate in the infinite width limit, the paper uses it to justify behaviors observed in the finite width limit. The justification for doing so is that the infinite width of the GCN does not affect the graph convolutions. However, infinite width certainly affects performance, as it corresponds to an overparameterized regime.
- The numerical experiments are limited. Specifically, they are restricted to class separability on SBMs and node classification on citation networks. Neither of these problems are adequate choices to analyze the effect of GCN depth, as both are relatively simple, consensus-like problems in which shallow and linear models have been shown to achieve very good performance. Additionally, the classification accuracy, which is the evaluation metric of interest for Cora, is not reported for this dataset.

Minor:

- The **Remark on Assumption 1**, particularly that "the linearized GCN performance is at par with the non-linear models with much reduced complexity", is misleading. This may be true in consensus-like problems like Cora and Citeseer, but it is not the case in problems where the relevant information lies in high graph frequencies. See, e.g., Fig. 3c in [r2].
- It is not "conventional" wisdom that GCNs "exhibit improvement in performance as depth increases" (third paragraph of the introduction). Overly increasing the depth may lead to overfitting.

[r1] Jeong, Sowon, and Claire Donnat. "Tuning the Geometry of Graph Neural Networks." arXiv preprint arXiv:2207.05887 (2022).
[r2] Ruiz, Luana, Luiz FO Chamon, and Alejandro Ribeiro. "Transferability Properties of Graph Neural Networks." arXiv preprint arXiv:2112.04629 (2021).

**Summary Of The Paper:**

This paper defines a graph neural tangent kernel (NTK) in a semi-supervised node classification setting. Under the degree-corrected stochastic block model (SBM), it then uses the graph NTK to explain, theoretically and empirically, why (i) row normalization performs better than symmetric normalization; (ii) with row normalization, performance degrades more slowly with the network depth; and (iii) skip connections improve performance for both types of normalization.

**Summary Of The Review:**

The problem addressed by the paper---understanding the effect of design choices such row/symmetric normalization, depth, and residual connections---is timely and important. However, the motivation for the proposed approach is ill-defined, the modeling assumptions restrict the applicability of the analysis, and the numerical experiments are limited.

---

> ### Author Response · Authors · 2022-11-17
> **Response to reviewer's concerns (motivation, known result, DC-SBM assumption, theorem 2, infinite width) [1/2]**
>
> The reviewer expresses concerns about our motivation since for a particular depth the performance difference between symmetric and row normalization is not significant. We disagree with this concern as experiments reveal the behavior for a very specific setting of the model and data. On the other hand, theory provides precise quantification that describes the behavior holistically. Hence, we think it is worthwhile to analyze this problem.
>
> We clarify the concerns raised by the reviewer in the following.
>
> ### 1. Motivation to study different convolutions and the role of depth
> Although the performance difference between symmetric and row normalization appears insignificant for depth=2 in Figure 1, [Wang et al.] observed a more significant difference of 3% for the same setup. This in itself shows the variability of experiments as it is only possible to observe certain characteristics in a specific setting (particular data, model, parameters, etc.). But, with rigorous theoretical analysis, we obtain the exact characterization of the learning problem under mild assumptions. Furthermore, the difference gets prominent with depth and thus it is as significant as understanding over smoothing in GCNs. Our analysis of the role of depth is complementary to the existing results on over smoothing as we used graph NTK to derive it.
>
> ### 2. Row normalization is better than symmetric normalization for community detection in general
> Although there exists traditional literature on this, it is not established when it comes to GCNs to the best of our knowledge. It needs a separate study in the case of GCNs, as learning of weights happens in each layer along with the convolution that may affect the behavior. Our work theoretically establishes this specifically for GCNs.
>
> ### 3. DC-SBM graph model assumption
> Heterophilous graphs are also included in our analysis since it can be modelled by taking $p<q$ in DC-SBM. Our initial experiments focused on homophilous graphs since GCNs are known to perform well in this case, but we have addressed this concern in the updated version of the paper by including new experiments that cover heterophilous graphs showing that the proposed approach and the results extend to this setting as well. Please refer to response 1 for reviewer *mpKm* for more detailed discussion.
>
> ### 4. Intuition and interpretation of Theorem 2
> The theorem gives intuition on the factors that affect the underlying class structure for a GCN of depth $d$ using the population NTK. For instance, in the case of symmetric normalization, the population NTK of depth $d$ shows that the class structure (in-class and out-of-class block difference) is affected by degree correction $\mathbf{\pi}$ and class separability $r$.
>
> To interpret the theorem, the basic idea is to analyze the convolutions on the grounds of how the underlying class structure of the DC-SBM ($p$ and $q$ blocks) is reflected in the population NTK. This implies that the presence of $\mathbf{\pi}$ in the expression is undesirable as it destroys information about the class structure. On this basis, only row normalization is unaffected by $\pi$ with the block difference proportional to $r=\frac{p-q}{p+q}$. Hence retaining the class structure implies the best among the convolutions. We have clarified this in the updated version of the paper.
>
> ### 5. Infinite width affects performance
> Although the infinite width affects the absolute accuracy, the trend in performance with different convolutions (row normalization performs better than symmetric normalization) matches the trained GCNs. This is because the convolutions do not operate on the width of the network, but rather on the nodes. So the absolute performance would change but the impact remains the same. This is supported by the experiments as well, thus making it suitable for our analysis. We added a comment in the introduction and an additional plot (Figure 5) to the appendix to further illustrate this relation empirically.
>
> ### References
> [Wang et al.] Attack graph convolutional networks by adding fake nodes. In Proceedings of Woodstock’18: ACM Symposium on Neural Gaze Detection, Woodstock, NY, 2018.

---

> > ### Author Response · Authors · 2022-11-17
> > **Response to reviewer's concerns (limited experiment, performance on cora and others) [2/2]**
> >
> > ### 6. Limited experiments
> > Since the focus of our paper is to theoretically analyze the empirical observations of GCN, we conducted experiments on the most evaluated datasets such as Cora and Citeseer in which GCN is known to perform well. As our focus is not on the absolute performance (measured in accuracy) but on the relative performance of the different diffusion operators our analysis still applies. Therefore, we have added experiments for heterophilic  (Figure 2) and core-periphery structure graphs (Section 5 and Figures 10, 11) in the updated version of the paper.
> >
> > ### 7. Classification accuracy on Cora
> > The accuracy on Cora dataset is plotted in Figures 1 and 5.
> >
> > ### 8. Minor comments
> > 1. **Remark on Assumption 1**: We agree with the reviewer that this may be true only for consensus-like problems as there exists no theoretical study on this. However, [Wu et al.] performed an extensive empirical study of linearized vs non-linear GCNs on a range of datasets and observed that the linear GCN performs at par with non-linear GCN. The comment made based on reference [r2] by the reviewer is unclear to us and we would like to get clarification on what the reviewer points to in the figure in [r2], and would be happy to address it.
> > 2. We would like to clarify that the comment made about the conventional wisdom is not about GCNs but about the standard deep neural networks. We clarified this in the revised version.
> >
> > ### 9. Clarity
> >
> > 1. **Assumption 2**: To compute the expected NTK, we need to consider random graph setting which results in an involved analysis. Hence, we considered $A=M$, which implies we assume the observed graph is its expectation and hence no randomness. In this setting, the computed NTK is simply the population version of it. Moreover, this assumption is commonly considered to analyze complex models under some distributional assumption, that are too complex to be analyzed in an expectation setting.
> > 2. **Theorem 2**: Although further simplification is not possible, the illustration of the theorem is given in the plots (row normalization preserving class structure as heatmaps and role of depth as line plots in Figure 2).  We have also added clarification to ‘Comparison of graph convolutions’ passage after the theorem in the updated version.
> >
> > ### 10. Correctness
> > We ask the reviewer to point us to the incorrect claims if they are not addressed above and we would be happy to address them.
> >
> > ### References
> > [Wu et al.] Simplifying Graph Convolutional Networks. International Conference on Machine Learning, Long Beach, California, PMLR 97, 2019.

---

> > > ### Comment · Reviewer_nvrd · 2022-11-21
> > > **Further concerns**
> > >
> > > Thank you for addressing my comments. I appreciate the clarifications and the inclusion of experiments on graphs other than homophilic graphs. However, I still have some concerns:
> > >
> > > - Assumption 1 is unclear to me. If $\mathbf{X} \in \mathbb{R}^{n \times f}$, shouldn't feature independence be expressed as $\mathbf{X}^{T}
> > >  \mathbf{X} = \mathbf{I}$ instead? If $\mathbf{X} \mathbf{X}^{T} = \mathbf{I}$ is indeed required, that is an extremely unrealistic assumption for a graph.
> > > - I appreciate the addition of the remark after Theorem 2, but the theorem is still difficult to parse. Additionally, I do not believe that the fact that $\pi$ does not appear in the NTK in the row-normalized case, but appears in the others, is sufficient to claim that the GSOs other than the row-normalized GSO have significantly worse class separability theoretically. All NTKs still depend on the SBM parameters $p$ and $q$. From this theorem, it is not obvious that the class information will be obscured just because the NTK also depends on $\pi$.
> > > - I still find the contribution of this paper of limited applicability. Node classification is only one type of node-level task, and it is not clear what these results would mean for other node-level tasks. This is where I believe empirical results on other datasets would help.

---

> > > > ### Author Response · Authors · 2022-11-22
> > > > **Clarification of the further concerns**
> > > >
> > > > We thank the reviewer for the comments. We provide clarification in the following,
> > > > ### 1. Assumption 1
> > > > The assumption on the features is made only to simplify the NTK expressions in Theorem 2. We can derive the same result without this assumption, for example by considering Contextual Stochastic Block Model [Deshpande et al., Ma et al.] which will lead to more complex NTK expressions. In Contextual SBM, the features of node $i$, $x_i \sim z_i \mu + \mathcal{N}(0, \sigma^2 I_f)$,  where  $\mu \in \mathbb{R}^f$ and $z_i = +1$ if node $i \in \mathcal{C}_1$, $-1$ if $i \in \mathcal{C}_2$. Under this consideration we can write the population version of $XX^T$ as $ z \mu^T \mu z^T = || \mu ||^2 zz^T$ where $z = (z_1, \ldots, z_n) \in \mathbb{R}^n$. Using this in the GNTK equation 4 in the paper, the analysis will result in the similar characterization as Theorem 2 with more complex terms, but the same conclusion about the convolution operators. \
> > > > On one hand, the reviewer questions the assumptions without which the NTKs in the theorem would be more complicated to express as discussed above, on the other hand, also raises concerns about the difficulty to parse the theorem in its current form. We would ask the reviewer to mention the parts of the NTK expressions that are difficult to parse so that we could think of ways to simplify them if possible.
> > > >
> > > > In addition, although this assumption on features is unrealistic in practice, it is **not restrictive** for the theoretical analysis, since the empirical observation that row normalization is better than symmetric normalization holds for both cases - with and without feature information. Moreover, we showed that the theoretical results hold in real datasets in both cases - Figure 12 in the appendix shows the evaluation of our result on Cora with the assumption $XX^T = I$, and Figure 4 in the main paper shows it without the assumption, that is, $XX^T \ne I$. Figures 13, 14 and 15 also illustrate the result without this assumption on real datasets.
> > > > We have also discussed this in the conclusion of the paper and further extensions of our work can be done by considering $XX^T \ne I$ to understand the interplay between features and graph information.
> > > >
> > > > ### 2. Further clarification on $\pi$ dependence
> > > > **Our interpretation of the comment:**
> > > >
> > > > The NTKs still have dependence on $p$ and $q$. So, the dependence on $\pi$ should not affect identifying the class structure for the graph convolutions. We ask the reviewer to correct our interpretation if mistaken.
> > > >
> > > > **Our response:**
> > > >
> > > > Although the NTKs have $p$ and $q$, the class structure gets diffused unevenly because of the presence of $\pi$ in all the convolutions except row normalization. This is illustrated in Figure 11 where we consider core-periphery model in each community, that is, $p>q$. As can be seen from the first plot of the model, the inhomogeneous degree correction acts as a noise partially obscuring the underlying class structure. The next two plots show the NTKs for symmetric normalization and row normalization of shallow depth $2$. One can see that the block structure is visible in first plot (DC-SBM), but gets more obscure in NTK for symmetric normalization. Row normalization, however, cleans the noise due to $\pi$ and clearly shows the communities (third plot). As demonstrated in this figure, row normalization effectively recovers the signal (block structures based on $p$ and $q$) and removes the noise (degree correction $\pi$) much better than symmetric normalization. Thus, independence of $\pi$ enables recovering the block structure perfectly given $p \ne q$ as characterized by Theorem 2. Hope this clarifies.
> > > >
> > > >
> > > > ### 3. Evaluation on other node-level tasks
> > > > We disagree with the reviewer on this point and we feel it is important to publish this work, as our analysis sets up a fundamental framework that can lead to follow-up works analyzing other node-level tasks. For instance, our analysis shows that row normalization is preferable for community detection even in heterophilic networks. As discussed in the conclusion of the paper, the analysis can further extend to link prediction as well. Moreover, node classification itself is one of the most important areas of research in graphs, and there exists no prior analysis of the graph convolutions. Hence, the presented theoretical analysis and the results are not limited in its applicability.
> > > >
> > > > ### References
> > > > [Deshpande et al.] Contextual stochastic block models. NeurIPS, 2018.
> > > >
> > > > [Ma et al.] Community Detection with Contextual Multilayer Networks. CoRR, abs/2104.02960.

---

### Official Review · Reviewer_mpKm · 2022-10-31

**Confidence:** 3
**Correctness:** 4
**Technical Novelty And Significance:** 4
**Empirical Novelty And Significance:** 3
**Recommendation:** 8

**Clarity, Quality, Novelty And Reproducibility:**

It is well written, well structured, and is of broad interest to an audience in theoretical machine learning on graphs. Their results are assumptions are clearly stated. The results are novel.
They cite other github repos such as GCN code they have adapted code from, but do not provide a link to their own code (as far as I can see). For reproducibility, I would suggest adding a github link to the authors' code in the NTK limit.

**Strength And Weaknesses:**

Strength: Results are clear, novel, and interesting, the math seems correct, and the paper brings a new perspectives on GNNs in the NTK limit.

Comments:
The results, although compelling, are restricted to one example, i.e., preserving class structure in one model of graphs (DC SBMs). I would be interested in seeing if this holds true more generally for other models of random graphs. For instance if one considered a graph with a few core nodes and some peripheral structure, is row normalization the best choice to preserve such ‘importance’ structure here as well (where one could look at the difference between core-to-core and core-to-periphery as a metric) – this should generally hold I think, since diffusion operators capture notions of ‘importance’? Classes here could be defined as communities analogous to the authors’ suggestion, but the degree distribution within each class is bimodal.
 How about graphs with many triangles, versus graphs that are locally tree-like? Does the analysis extend just as easily to signed graphs (positive and negative weights)? Even if not, a discussion on assumptions of graph structure considered here would be very useful.
A question of particular interest to me (which may be beyond the scope of this paper), is if there exists a normalization (perhaps one of the four proposed) that best preserves the ‘topological’ structure of the graph, i.e., holes in graph, in addition to the class structure.

Could one define a metric for ‘distance’ between graphs. One might define two graphs to be ‘structurally’ similar if the have equivalent ‘mixing’ between classes after d layers. A GNN with improved expressive power should be better for defining better distance metrics, I think.

What is the conclusion from the section on skip connections? Is one type of skip connection supposed to work better with row norm, and another with sym? This was not clear to me. Panels on second row of Fig 4 (comparison between skip-PC and skip-alpha) look very similar to me.


**Summary Of The Paper:**

The paper investigates the effects of different normalizations of the adjacency matrix as the convolution operator in graph neural networks. In particular, they present a theoretical investigation of GNNs in the Neural Tangent Kernel (NTK) limit (very wide limit where training the GNN is equivalent to kernel regression on the NTK – without the need to optimize hyperparameters). They study the effects of four different convolution operators with different normalizations on SBMs, and show that some normalizations (row norm) preserve the ‘block’ structure better than others even at (relatively) large depths of 5 or more. They show empirically that row normalization is the most likely to preserve this structure, while other operators diffuse information over the graph as the network gets deeper. The authors also investigate two types of skip connections and show theoretically that they retain the structural information of the SBM even at infinite width.

**Summary Of The Review:**

The paper presents an interesting study on the expressive power of GNNs. It presents theoretical results (under certain orthonormality of features conditions), but shows that some of the results hold empirically in a more general case.
Graph NTKs have been introduced before…the novelty of this work is in the study of different normalizations of the convolution operator. Overall, while I have some questions on the generality of their results to other random graph models (if the results are general, this would strengthen their results), I believe the results are informative and interesting, and would recommend for publication, especially with additional discussion and testing for generality.

---

> ### Author Response · Authors · 2022-11-17
> **Response to reviewer's comments (extending the analysis to different models) [1/2]**
>
> We thank the reviewer for the positive and constructive feedback. We address the given comments below and include them as much as possible into the revised version of the paper.
> ### 1. Extending the analysis to different graph models
> Thanks for raising this point. Although our analysis is currently done on DC-SBM, it serves as a framework for studying any random graph model. Initially we focused on homophilous graphs modeled by DC-SBM since GCNs are known to perform well in this case and need architectural changes otherwise [Zheng et al]. However, the analysis for graphs with heterophily and core-periphery structures can be incorporated in the current analysis by doing the following:
>
> * **Heterophily graphs**: our analysis does not assume any condition on the class linking probabilities $p$ and $q$ of DC-SBM. Therefore, we can model heterophilic graphs by considering $0 \leq p < q \leq 1$ (homophilic graphs: $0 \leq q < p \leq 1$). The conclusion on convolution operators holds as well. Refer to the new experiment added to Figure 2. By considering the right column we clearly see that the ordering of convolutions with respect to the preservation of the average gap between in-class and out-of-class blocks is the same for both the homophilic and heterophilic cases. The presented heatmaps further support this result.
> * **Graphs with core-periphery structure**: Here, we can have two cases, where both cases can be modeled by DC-SBM and therefore directly follow from Theorem 2:
>      1. each class has core-periphery structure, and core-core connects strongly between classes compared to core-periphery and periphery-periphery between classes. In this case, the analysis directly follows analogous to the homophilic case and the main findings on the ordering of convolution operators remain the same (see Figure 11 for illustration)
>      2. graphs with no underlying class structure $(p = q)$ but core-periphery structure where the graph has core nodes that are highly interconnected and periphery nodes that are sparsely connected to the core and other periphery nodes. In this case, we show that the convolution $S_{sym}$ preserves the graph information from Theorem 2. We formalize this result in Corollary 4 and illustrate in Figure 10. This result is not surprising as it directly follows from Theorem 2 when $p=q$.
>
> We have included the discussion on modelling the above graphs in the ‘Random Graph Model’ passage of the paper in Section 3 and added additional experiments (Figures 2, 10, 11) demonstrating the theory in homophilic, heterophilic and core-periphery cases for different depths.
>
> To comment on the other cases mentioned by the reviewer,
>
> * **Signed graphs**: Signed graphs can be modeled by Signed Stochastic Block Model [Yang et al]. In expectation, there is fundamentally no difference between SSBM and SBM, which is also well established in community detection [Yun et al]. Hence, we do not expect the results to change for signed graphs even when random graphs are considered.
>  * **Graphs with many triangles vs tree-like**: in these cases, GNTK analysis based on Theorem 1 extends easily as shown for other graphs, but the representation power of  different convolutions needs to be studied separately and a conclusive analysis of these graphs is beyond the scope of the current work.
> * **Topological structure preservation**: this is an interesting direction, however, we cannot comment on the normalization without further analysis. The general NTK framework can be used in this case also (see a more detailed discussion below).
> We have added these possibilities in the Conclusion of the paper.
>
> ### 2. Analysis of normalization that preserves the topological structure in graphs
> This setting is slightly different in the sense that the focus is on preserving the graph structure itself in comparison to preserving the class structure. However, starting from the NTK formulation presented in Theorem 1 an analysis could be possible. Essentially one would have to compute the measure of topological structure on the original graph (most likely some measure of simplicial complexes), then perform the transformation using the NTK from Theorem 1, and then again look at the structural measure (this should be possible as the GNTK provides a simple algebraic transformation). A more thorough analysis of this setting is beyond the scope of this rebuttal but most likely possible with the presented overall approach.
>
> ### References
> [Zheng et al] Graph Neural Networks for Graphs with Heterophily: A Survey. arxiv:2202.07082 (2022)
>
> [Yang et al] Bayesian Approach to Modeling and Detecting Communities in Signed Network. AAAI (2015)
>
> [Yun et al] Optimal Cluster Recovery in the Labeled Stochastic Block Model. NeurIPS (2016).

---

> > ### Author Response · Authors · 2022-11-17
> > **Response to reviewer's comments (distance metric, conclusion from skip connection, reproducibility) [2/2]**
> >
> > ### 3. Distance metric for graphs
> > While our analysis shows that row normalization has better representation power, we think it may *not* be suitable for defining distance metric between graphs. The GNTKs for two graphs sampled from different models could structurally be similar. For instance, in the updated Figure 2, row normalization GNTKs of depth=2 of heterophily and homophily graphs are similar which shows the distance metric won’t be suitable for comparing different graphs.
> >
> > ### 4. Conclusion from skip connection
> > Our conclusion here is that the block difference between in-class and out-of-class is retained even for deeper GCNs with skip connections. While we do not compare the two skip connections, our objective is to include the analysis of the impact of convolving with and without the feature information in the skip connections as well, and hence we chose Skip-PC where the skip connection is added to the features before convolving and Skip-$\alpha$ where the features are added to each layer without convolving. The analysis shows that the result remains the same in both cases. We have clarified this in the updated version of the paper. Finally, we note that the results for skip connections extend to graphs with homophilic, heterophilic and core-periphery structures as well.
> >
> > ### 5. Reproducibility
> > The code is provided as a zip file in the supplementary material and we will provide the link to github in the final camera-ready version due to the anonymity concerns.

---

### Author Response · Authors · 2022-11-17
**Addressing a common concern from reviewers and updates done to the paper**

We would like to commonly address a concern from reviewers **nvrd**, **aSCc** and **vkDP** on the significance of the work and mention the updates done to the paper to address the review comments.

Quoting from the reviews:

**nvrd**: `..if the best accuracies in this figure are achieved for 2 layers, and the difference between symmetric and row normalization in the 2-layer case is small, why are we concerned with what happens for a larger (or even infinite) number of layers?`

**aSCc**: `... the superiority of row-normalized adjacency matrices is not well-known.`

**vkDP**: `No further proposal is made based on the proof. It lacks a further discussion on what we can do differently from the current existing approaches.`

All three reviewers seem to suggest that there is no point in the theoretical analysis done in this paper. However, we note that precise theoretical characterization of deep neural networks is a significant part of machine learning research, and ICLR publishes such papers [Geerts et al, Wei et al]. Hence, we believe that the present work is relevant to the ICLR community, even though it does not propose new methods.

The reviewers question the motivation of the work since empirically the superiority of row normalization is not well known or the difference in performance seems insignificant. However, we disagree with this being a drawback as symmetric normalization is widely used in practice because it was suggested in popular GCN works and the study of different convolutions in the context of GCNs is not theoretically derived. Moreover, the widely adopted pytorch implementation of GCN (https://github.com/tkipf/pygcn/blob/master/pygcn/utils.py#L39) uses row normalization as default while the paper suggests symmetric normalization. (See response to reviewer *aSCc* for more details). In fact, given this and the theoretical results from our work, there is a possibility that some of the works assume symmetric normalization but actually use row normalization in practice, and also the potential of significant improvement in the performance if symmetric normalization is replaced with row normalization in the extensive experiments conducted using GCNs so far. To conclude, **we focus on the theory to give precise characterization** which we feel is relevant for ICLR readers.

### Updates to the paper
To strengthen the significance of the contribution, we show how the given main result in Theorem 2 applies to graphs with *homophilic, heterophilic and core-periphery structures*. We note that no new theorems were derived, but special cases of Theorem 2 are presented in the updated Figure 2 and the new section 5. We briefly mention the major updates done to the paper:

1.  Generalizing the analysis: Major comment from *mpKm* and *nvrd* is to analyze beyond (homophilic) community-like structures in the graph. Our current analysis framework includes different graph structures like homophilic, heterophilic and core-periphery.
     * Theoretical discussions in Section 3 (In section ‘Random Graph Model.’ we add a discussion of how homophilic graphs, heterophilic graphs and core-periphery graphs can be modelled using DC-SBM.) and Section 5 (Analysis of core-periphery graphs).
     * New experimental results for homophilic, heterophilic and core-periphery structures in Figures 2, 10, 11.
     * Relevant changes in abstract, introduction and conclusion
2.  We add Figure 5 to the appendix illustrating that the NTK behaves similarly to GCN in trends with depth and discuss in the introduction addressing comment from *nvrd*.
3.  Figure 4 is updated with results using linear GCN and moved the relu GCN results to Appendix B.4 (Figure 13) to address the concern on relationship between the theory and numerics from reviewer *aSCc*.
4. Typo in skip-PC and Skip-$\alpha$ equations in Section 4.2 is corrected.
5.  Remaining changes are formatting to fit to 9 pages.

We mistook the discussion period till November 18th as the time to update the paper and the discussions with the reviewers to happen even in stage 2 since the ‘Reviewing process’ in the ICLR webpage states it slightly differently, hence the delay. So, we ask the reviewers to kindly consider the updated version for further discussion. Thanks.

### References
[Geerts et al] Expressiveness and Approximation Properties of Graph Neural Networks, ICLR 2022 (Oral)

[Wei et al] Theoretical Analysis of Self-Training with Deep Networks on Unlabeled Data, ICLR 2021 (Oral)

---

### Decision · Program_Chairs · 2023-01-20

**Decision:**

Reject

**Justification For Why Not Higher Score:**

As stated above, the main criticisms expressed by the reviewers were (a) a lack of justification for the study of the normalized and symmetric convolutions through the degree corrected SBM, (b) a lack of appropriate numerical experiments, and (c) the inability to generalize to other graphs beyond the degree corrected SBM that the authors consider. While during the review process the authors have attempted to address these issues by running experiments on other types of graphs, the edits do not seem to have satisfied fully the reviewers.

**Justification For Why Not Lower Score:**

N/A

**Metareview: Summary, Strengths And Weaknesses:**

__Summary.__ This work proposes an analysis of the impact of the convolution operator on the performance of GNN through the lens of Neural Tangent Kernels for semi-supervised node classification tasks. Under the degree-corrected stochastic block model (SBM), it uses the graph NTK to explain, theoretically and empirically, why  and when row normalization performs better than symmetric normalization; They further show that, with row normalization, performance degrades more slowly with the network depth; and finally, they highlight why skip connections improve performance for both types of normalization. The use of the NTK foregoes the problem of parameter tuning, and allows them to draw insights on the benefits of the different types of convolutions.


__Overview of criticisms raised during the review process.__
Overall,  at the exception of one reviewer who deems the paper to be good, all the other three vote for a strong rejection (note that we are excluding the opinion of a fifth reviewer, who declared themselves unable to assess the paper). The main criticisms expressed by the reviewers were (a) a lack of justification for the study of the normalized and symmetric convolutions through the degree corrected SBM, (b) a lack of appropriate numerical experiments, and (c) the inability to generalize to other graphs beyond the degree corrected SBM that the authors consider. While during the review process the authors have attempted to address these issues by running experiments on other types of graphs, the edits do not seem to have satisfied fully the reviewers. The consensus amongst the reviewers seems to be that the results are hard to interpret as well. The approach was however interesting and the problem, “timely” as expressed by the reviewers. __While we advise a rejection of this paper, we therefore encourage the authors to continue improving their paper based on the feedback that they have received and re-submit in the near future.__


**Summary Of Ac-Reviewer Meeting:**

N/A